# Meta-Learning Priors for Safe Bayesian Optimization

**Jonas Rothfuss**
ETH Zurich
Switzerland
rojonas@ethz.ch

**Christopher Koenig**
inspire AG, ETH
Switzerland
chkoenig@ethz.ch

**Alisa Rupenyan**
inspire AG, ETH
Switzerland
ralisa@ethz.ch

**Andreas Krause**
ETH Zurich
Switzerland
krausea@ethz.ch

**Abstract:** In robotics, optimizing controller parameters under safety constraints is an important challenge. Safe Bayesian optimization (BO) quantifies uncertainty in the objective and constraints to safely guide exploration in such settings. Hand-designing a suitable probabilistic model can be challenging however. In the presence of unknown safety constraints, it is crucial to choose reliable model hyper-parameters to avoid safety violations. Here, we propose a data-driven approach to this problem by *meta-learning priors* for safe BO from offline data. We build on a meta-learning algorithm, F-PACOH, capable of providing reliable uncertainty quantification in settings of data scarcity. As core contribution, we develop a novel framework for choosing *safety-compliant priors* in a data-riven manner via empirical uncertainty metrics and a *frontier search* algorithm. On benchmark functions and a high-precision motion system, we demonstrate that our meta-learned priors accelerate the convergence of safe BO approaches while maintaining safety.

**Keywords:** Meta-learning, Safety, Controller tuning, Bayesian Optimization

## 1  Introduction

Optimizing a black-box function with as few queries as possible is a ubiquitous problem in science and engineering. Bayesian Optimization (BO) is a promising paradigm, which learns a probabilistic surrogate model (often a Gaussian process, GP) of the unknown function to guide exploration. BO has been successfully applied for optimizing sensor configurations [1, 2] or tuning the parameters of robotic controllers [3, 4, 5, 6, 7]. However, such real-world applications are often subject to safety constraints which must not be violated in the process of optimization, e.g., the robot not getting damaged. Often, the dependence of the safety constraints on the query inputs is a priori unknown and can only be observed by measurement. To cope with these requirements, safe BO methods [8, 9, 10] model both the objective *and constraint* functions with GPs. The uncertainty in the constraint is used to approximate the feasible region from within, to guarantee that no safety violations occur. Therefore, a critical requirement for safe BO is the reliability of the uncertainty estimates of our models. Typically, it is assumed that a correct GP prior, upon which the uncertainty estimates hinge, is exogenously given [e.g., 8, 9, 10]. In practice, however, appropriate choices for the kernel variance and lengthscale are unknown and typically have to be chosen very conservatively or hand-tuned by trial and error, a problematic endeavour in a safety-critical setting. A too conservative choice dramatically reduces sample efficiency, whereas overestimating smoothness may risk safety violations.

Addressing these shortcomings, we develop an approach for *meta-learning informative, but safe, GP priors in a data-driven way* from related offline data. We build on the F-PACOH meta-learning method [11] which is capable of providing reliable uncertainty estimates, even in the face of data-scarcity and out-of-distribution data. However, their approach still relies on an appropriate kernel choice on which it falls back in the absence of sufficient data. We propose a novel framework for choosing safety-compliant kernel hyper-parameters in a data-driven manner based on calibration and sharpness metrics of the confidence intervals. To optimize these uncertainty metrics, we devise a *frontier search* algorithm that efficiently exploits the monotone structure of the problem. The resulting *Safe Meta-Bayesian Optimization* (SAMBO) approach can be instantiated with existing safe BO

6th Conference on Robot Learning (CoRL 2022), Auckland, New Zealand.

methods and utilize the improved, meta-learned GPs to perform safe optimization more efficiently. In our experiments, we evaluate and compare our proposed approach on benchmark functions as well as controller tuning for a high-precision motion system. Throughout, SAMBO significantly improves the query efficiency of popular safe BO methods, without compromising safety.

## 2 Related Work

**Safe BO** aims to efficiently optimize a black-box function under safety-critical conditions, where unknown safety constraints must not be violated. Constrained variants of standard BO methods [12, 13, 14] return feasible solutions, but do not reliably exclude unsafe queries. In contrast, SAFEOPT [8, 9] and related variants [15] guarantee safety at all times and have been used to safely tune controllers in various applications [e.g., 16, 17]. While SAFEOPT explores in an undirected manner, GOOSE [10, 18] does so by expanding the safe set in a goal-oriented fashion. All mentioned methods rely on GPs to model the target and constraint function and assume that correct kernel hyper-parameters are given. Our work is complementary: We show how to use related offline data to obtain informative and safe GP priors in a data-driven way that makes the downstream safe BO more query efficient.

**Meta-Learning.** Common approaches in meta-learning amortize inference [19, 20, 21], learn a shared embedding space [22, 23, 24, 25] or a good neural network initialization [26, 27, 28, 29]. However, when the available data is limited, these approaches a prone to overfit on the meta-level. A body of work studies meta-regularization to prevent overfitting in meta-learning [30, 31, 32, 33, 34, 35]. Such meta-regularization methods prevent meta-overfitting for the mean predictions, but not for the uncertainty estimates. Recent meta-learning methods aim at providing reliable confidence intervals even when data is scarce and non-i.i.d [11, 36, 37]. These methods extend meta-learning to interactive and life-long settings. However, they either make unrealistic model assumptions or hinge on hyper-parameters whose improper choice is critical in safety constraint settings. Our work uses F-PACOH [11] to meta-learn reliable GP priors, but removes the need for hand-specifying a correct hyper-prior by choosing its parameters in a data-driven, safety-aware manner.

## 3 Problem Statement and Background

### 3.1 Problem Statement

We consider the problem of safe Bayesian Optimization (safe BO), seeking the global minimizer

$$\mathbf{x}^* = \arg\min_{\mathbf{x} \in \mathcal{X}} f(\mathbf{x}) \quad \text{s.t.} \quad q(\mathbf{x}) \leq 0 \tag{1}$$

of a function $f : \mathcal{X} \to \mathbb{R}$ over a bounded domain $\mathcal{X}$ subject to a safety constraint $q(\mathbf{x}) \leq 0$ with constraint function $q : \mathcal{X} \to \mathbb{R}$. For instance, we may want to optimize the controller parameter of a robot without subjecting it to potentially damaging vibrations or collisions. During the optimization, we iteratively make queries $\mathbf{x}_1, ..., \mathbf{x}_T \in \mathcal{X}$ and observe noisy feedback $\tilde{f}_1, ..., \tilde{f}_T$ and $\tilde{q}_1, ..., \tilde{q}_T$, e.g., via $\tilde{f}_t = f(\mathbf{x}_t) + \epsilon_f$ and $\tilde{q}_t = q(\mathbf{x}_t) + \epsilon_q$ where $\epsilon_f, \epsilon_q$ is $\sigma^2$ sub-Gaussian noise [38, 39]. In our setting, performing a query is assumed to be costly, e.g., running the robot and observing relevant measurements. Hence, we want to find a solution as close to the global minimum in as few iterations as possible without violating the safety constraint with any query we make.

Additionally, we assume to have access to datasets $\mathcal{D}_{1,T_1}, ..., \mathcal{D}_{n,T_n}$ with observations from $n$ statistically similar but distinct data-generating systems, e.g., data of the same robotic platform under different conditions. Each dataset $\mathcal{D}_{i,T_i} = \{(\mathbf{x}_{i,1}, \tilde{f}_{i,1}, \tilde{q}_{i,1}), ..., (\mathbf{x}_{i,T_i}, \tilde{f}_{i,T_i}, \tilde{q}_{i,T_i})\}$ consists of $T_i$ measurement triples, where $\tilde{f}_{i,t} = f_i(\mathbf{x}_{i,t}) + \epsilon_{f_i}$ and $\tilde{q}_{i,t} = q_i(\mathbf{x}_{i,t}) + \epsilon_{q_i}$ are the noisy target and constraints observations. We assume that the underlying functions $f_1, ..., f_n$ and $q_1, ..., q_n$ which generated the data are representative of our current target and constraint functions $f$ and $q$, e.g., that they are all i.i.d. draws from the same stochastic process. However, the data within each dataset may be highly dependent (i.e., non-i.i.d.). For instance, each $\mathcal{D}_{i,T_i}$ may correspond to the queries and observations from previous safe BO sessions with the same robot under different conditions.

In this paper, we ask the question of how we can harness such related data sources to make safe BO on our current problem of interest more query efficient, without compromising safety.

## 3.2 Safe Bayesian Optimization Methods

Safe BO methods construct a Bayesian *surrogate model* of the functions $f$ and $q$ based on previous observations $\mathcal{D}_t = \{(\mathbf{x}_{t'}, \tilde{f}_{t'}, \tilde{q}_{t'})\}_{t' < t}$. Typically, a Gaussian Process GP $(f(\mathbf{x})|m(\mathbf{x}), k(\mathbf{x}, \mathbf{x}'))$ with mean $m(\mathbf{x})$ and kernel function $k(\mathbf{x}, \mathbf{x}')$ is employed to form posterior beliefs $p(f(\mathbf{x})|\mathcal{D}_t) = \mathcal{N}(\mu_t^f(\mathbf{x}), (\sigma_t^f(\mathbf{x}))^2)$ and $p(q(\mathbf{x})|\mathcal{D}_t) = \mathcal{N}(\mu_t^q(\mathbf{x}), (\sigma_t^q(\mathbf{x}))^2)$ over function values [40]. Based on the predictive posterior, we can form confidence intervals (CIs) to the confidence level $\alpha \in [0, 1]$

$$CI_\alpha^f(\mathbf{x}|\mathcal{D}_t) := \left[\mu_t^f(\mathbf{x}) \pm \beta^f(\alpha)\sigma_t^f(\mathbf{x})\right] \tag{2}$$

where $\beta^f(\alpha)$, the scaling of the standard deviation, is set such that $f(x)$ is in the CI with probability $\alpha$. For BO, we often employ a shift invariant kernel $k(\mathbf{x}, \mathbf{x}') = \nu\phi(||\mathbf{x} - \mathbf{x}'||/l)$, where $\nu$ is its variance, $l$ the lengthscale and $\phi$ a positive function, e.g., squared exponential (SE) $\phi(z) = \exp(-z^2)$.

BO methods typically choose their query points by maximizing an acquisition function based on $p(f(\mathbf{x})|\mathcal{D}_t)$, trading-off exploration and exploitation [41, 42, 43, 44]. When we have safety constraints, we need to maintain a *safe set* $\mathcal{S}_t(\alpha) = \{\mathbf{x} \in \mathcal{X}|\mu_t^q(\mathbf{x}) + \beta^q(\alpha)\sigma_t^q(\mathbf{x}) < 0\}$ which contains parts of the domain we know to be safe with high-probability $\alpha$. To maintain safety, we can only query points within the current $\mathcal{S}_t(\alpha)$. In addition, we need to explore w.r.t. $q$ so that we can expand $\mathcal{S}_t$. E.g., SAFEOPT [8, 9, 16] computes a safe query candidate that optimizes the acquisition function for $f$ as well as a query candidate that promises the best expansion of $\mathcal{S}_t$. Then, it selects the one with the highest uncertainty. While SAFEOPT expands $\mathcal{S}_t$ undirectedly, GOOSE [10, 18] does so in a more directed manner to avoid unnecessary expansion queries, irrelevant for minimizing $f$. In addition to the pessimistic $\mathcal{S}_t$, it also maintains an optimistic safe set which is used to calculate a query candidate $\mathbf{x}_t^{\text{opt}}$ that maximizes the acquisition function for $f$. If $\mathbf{x}_t^{\text{opt}}$ is outside of $\mathcal{S}_t$, it chooses safe query points aiming at expanding $\mathcal{S}_t$ in the direction of $\mathbf{x}_t^{\text{opt}}$. See Appx. A for more details.

## 3.3 Meta-Learning GP Priors

In meta-learning [45, 46], we aim to extract prior knowledge (i.e., inductive bias) from a set of related learning tasks. Typically, the meta-learner is given such learning tasks in the form of $n$ datasets $\mathcal{D}_{1,T_1}, ..., \mathcal{D}_{n,T_n}$ with $\mathcal{D}_{i,T_i} = \{(\mathbf{x}_{i,t}, y_{i,t})\}_{t=1}^{T_i}$ and outputs a refined prior distribution or hypothesis space which can then be used to accelerate inference on a new, but related, learning task.

Prior work proposes to meta-learning GP priors [47, 48, 34], tough, fails to maintain reliable uncertainty estimates when data is scarce and/or non-i.i.d.. The recently introduced F-PACOH method [11] overcomes this issue, by using a regularization approach in the function space. As previous work [e.g., 49, 48], they use a learnable GP prior $\rho_\theta(h) = \text{GP}(h(\mathbf{x})|m_\theta(\mathbf{x}), k_\theta(\mathbf{x}, \mathbf{x}'))$ where the mean and kernel function are neural networks with parameters $\theta$ and employ the marginal log-likelihood to fit the $\rho_\theta(h)$ to the meta-training data. However, during the meta-learning, they regularize $\rho_\theta(h)$ towards a Vanilla GP hyper-prior $\rho(h) = \text{GP}(h(\mathbf{x})|0, k(\mathbf{x}, \mathbf{x}'))$ with the SE kernel. To do so, they uniformly sample random measurement sets $\mathbf{X} = [\mathbf{x}_1, ..., \mathbf{x}_m] \overset{\text{i.i.d.}}{\sim} \mathcal{U}(\mathcal{X})$ from the domain and compare the GPs finite marginals $\rho_\theta(\mathbf{h^X}) = p_\theta(h(\mathbf{x}_1), ...., h(\mathbf{x}_m))$ and $\rho(\mathbf{h^X})$ through their KL-divergence. The resulting meta-learning loss with the functional KL regularizer

$$\mathcal{L}(\boldsymbol{\theta}) := \frac{1}{n}\sum_{i=1}^{n}\left(-\frac{1}{T_i}\underbrace{\ln Z(\mathcal{D}_{i,T_i}, \rho_\theta)}_{\text{marginal log-likelihood}} + \left(\frac{1}{\sqrt{n}} + \frac{1}{nT_i}\right)\underbrace{\mathbb{E}_\mathbf{X}\left[KL[\rho_\theta(\mathbf{h^X})||\rho(\mathbf{h^X})]\right]}_{\text{functional KL-divergence}}\right) \tag{3}$$

makes sure that, in the absence of sufficient meta-training data, the learned GP behaves like a Vanilla GP. Overall, this allows us to meta-learn a more informative GP prior which still yields reliable confidence intervals, even if the meta-training data was collected via safe BO and is thus not i.i.d.

# 4 Choosing the Safe Kernel Hyper-Parameters

Important for safe BO is the reliability of the uncertainty estimates of our objective and constraint models. For GPs, the kernel hyper-parameters with the biggest influence on the CIs. If the kernel variance $\nu$ is chosen too small and/or the lengthscale $l$ to large, our models become over-confident

and the corresponding BO unsafe. In the reverse case, the CIs become too conservative and safe BO requires many queries to progress. Despite the assumption commonly made in earlier work, [e.g., 8, 9, 10], appropriate choices for $\nu$ and $l$ are unknown. In practice, they are typically chosen conservatively or hand-tuned by trial and error, problematic in safety-critical settings. Aiming to address this issue, we develop a framework for choosing the kernel hyper-parameters in a data-driven manner.

## 4.1 Assessing kernel hyper-parameters: Calibration and sharpness

Our approach is based on the *calibration* and *sharpness* of uncertainty estimates [see e.g. 50, 51, 52, 53]. Naturally, if we construct CIs to the confidence level $\alpha$, we want that at least an $\alpha$ percentage of (unseen) observations to fall within these CIs. If this holds in expectation, we say that the uncertainty estimates are *calibrated*. If the empirical percentage is less than $\alpha$, it indicates that our model's uncertainty estimates are over-confident and we are likely to underestimate the risk of safety violations. To empirically assess how calibrated a probabilistic regression model with hyper-parameters $\boldsymbol{\omega}$, conditioned on a training dataset $\mathcal{D}^{\mathrm{tr}}$, is, we compute its *calibration frequency* on a test dataset $\mathcal{D}^{\mathrm{test}}$:

$$\text{calib-freq}(\mathcal{D}^{\mathrm{tr}}, \mathcal{D}^{\mathrm{test}}, \boldsymbol{\omega}) := \frac{1}{|A|} \sum_{\alpha \in A} \mathbb{1} \left( \frac{1}{|\mathcal{D}^{\mathrm{test}}|} \sum_{(\mathbf{x},y) \in \mathcal{D}^{\mathrm{test}}} \left[ \mathbb{1} \left( y \in CI_\alpha^f \left( \mathbf{x} | \mathcal{D}^{\mathrm{tr}}, \boldsymbol{\omega} \right) \right) \right] \geq \alpha \right) . \quad (4)$$

Here, $A \subset [0,1]$ is a set of relevant confidence levels (in our case 20 values equally spaced between $0.8$ and $1.0$). Since the CIs of our model need to be calibrated at any iteration $t$ during the BO and for any task we may face, we choose the best empirical estimate we can. We compute the average calibration frequency across all meta-training datasets and for any sub-sequence of points within a dataset. In particular, for any task $i = 1, ..., n$ and $t = 1, ..., T_i - 1$ we condition/train our model on the data points $\mathcal{D}_{i, \leq t} = \{(\mathbf{x}_{i,t'}, y_{i,t'})\}_{t'=1}^t$ and use the remaining data points $\mathcal{D}_{i, >t} = \{(\mathbf{x}_{i,t'}, y_{i,t'})\}_{t'=t+1}^{T_i}$ to compute the calibration frequency. Overall, this gives us

$$\text{avg-calib}(\{\mathcal{D}_{i,T_i}\}_{i=1}^n, \boldsymbol{\omega}) := \frac{1}{n} \sum_{i=1}^n \frac{1}{T_i - 1} \sum_{t=1}^{T_i - 1} \text{calib-freq}(\mathcal{D}_{i, \leq t}, \mathcal{D}_{i, >t}, \boldsymbol{\omega}) . \quad (5)$$

While the calibration captures how *reliable* the uncertainty estimates are, it does not reflect how *useful* the confidence intervals are for narrowing down the range of possible function values. For instance, a predictor that always outputs a mean with sufficiently wide confidence intervals is calibrated, but useless for BO. Hence, we also consider the *sharpness* of the uncertainty estimates which we empirically quantify through the *average predictive standard deviation*. Similar to (5), we average over all tasks and data sub-sequences within each task:

$$\text{avg-std}(\{\mathcal{D}_{i,T_i}\}_{i=1}^n, \boldsymbol{\omega}) := \frac{1}{n} \sum_{i=1}^n \frac{1}{T_i - 1} \sum_{t=1}^{T_i - 1} \frac{1}{|\mathcal{D}_{i, >t}|} \sum_{(\mathbf{x},y) \in \mathcal{D}_{i, >t}} \sigma(\mathbf{x} | \mathcal{D}_{i, \leq t}, \boldsymbol{\omega}) . \quad (6)$$

The avg-std measures how concentrated the uncertainty estimates are and, thus, constitutes a natural complement to calibration which can be simply achieved by wide/loose confidence intervals.

## 4.2 Choosing good hyper-parameters via Frontier search

Based on the two empirical quantities introduced above, we can optimize the hyper-parameters $\boldsymbol{\omega}$ of our model as to maximize sharpness (i.e., minimize the avg-std) subject to calibration [50]:

$$\min_{\boldsymbol{\omega}} \ \text{avg-std}(\{\mathcal{D}_{i,T_i}\}_{i=1}^n, \boldsymbol{\omega}) \quad \text{s.t.} \ \text{avg-calib}(\{\mathcal{D}_{i,T_i}\}_{i=1}^n, \boldsymbol{\omega}) \geq 1 \quad (7)$$

Since computing $\text{avg-std}(\{\mathcal{D}_{i,T_i}\}_{i=1}^n, \boldsymbol{\omega})$ and $\text{avg-calib}(\{\mathcal{D}_{i,T_i}\}_{i=1}^n, \boldsymbol{\omega})$ requires solving the the GP inference problem many times, each query is computationally demanding. Hence, we need an optimization algorithm for (7) that requires as few queries as possible to get close to the optimal solution.

We develop an efficient *frontier search (FS)* algorithm that exploits the monotonicity properties of this optimization problem. Both avg-std and avg-calib are monotonically increasing in the kernel variance $\nu$ and decreasing in the lengthscale $l$[1]. By setting $\mathbf{z} = (-l, \nu)$ and writing

---

[1]Note that the monotonicity of the calibation frequency in $l$ is only an empirical heuristic that holds in almost all cases if $\nu$ is at least as big as the variance of the targets $y$ in a dataset.

**Algorithm 1** FRONTIERSEARCH (details in Appendix C)

**Input:** Domain bounds $\mathbf{z}^l, \mathbf{z}^u$ s.t. $\mathbf{z}^l \leq \mathbf{z}^* \leq \mathbf{z}^u$
1: $\mathcal{Q}^u \leftarrow \{\mathbf{z}^u\}, \mathcal{Q}^l \leftarrow \{\mathbf{z}^l\}$
2: **for** $k = 1, ..., K$ **do**
3:     $(\mathbf{z}_r, \mathbf{z}'_r) \leftarrow$ LARGESTMAXMINRECT$(\mathcal{Q}^l, \mathcal{Q}^u)$         // Largest max-min rect betw. frontiers
4:     $\mathbf{z}_q \leftarrow$ BESTWORSTCASEQUERY$(\mathbf{z}_r, \mathbf{z}'_r, \mathcal{Q}^l, \mathcal{Q}^u)$     // Best query point to split rectangle
5:     **if** $c(\mathbf{z}_q) \geq 1$ **then**
6:         $\mathcal{Q}_u \leftarrow$ PRUNE$(\mathcal{Q}^u \cup \{\mathbf{z}_q\})$
7:     **else**
8:         $\mathcal{Q}_l \leftarrow$ PRUNE$(\mathcal{Q}^l \cup \{\mathbf{z}_q\})$
**Return:** $\arg\min_{\mathbf{z} \in \mathcal{Q}^u} s(\mathbf{z})$

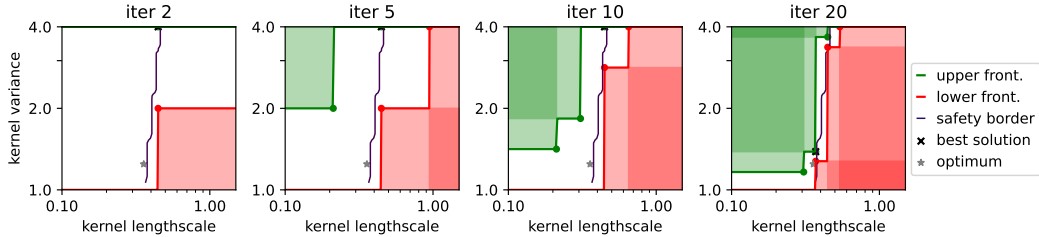

Figure 1: Frontier search (FS) on the kernel lengthscale and variance for the constraint model Argus robot. Red: areas ruled out, because unsafe. Green: Safe areas that are ruled out since dominated by better safe queries. After a few iterations, FS has already shrunk the set of possible optima (white area between fronts) to points close to the safety boarder and picked nearly optimal solution (cross).

$s(\mathbf{z}) = \text{avg-std}(\{\mathcal{D}_{i,T_i}\}_{i=1}^n, l, \nu)$ and $c(\mathbf{z}) = \text{avg-calib}(\{\mathcal{D}_{i,T_i}\}_{i=1}^n, l, \nu)$, we can turn (7) into

$$\min_{\mathbf{z}} \ s(\mathbf{z}) \quad \text{s.t.} \quad c(\mathbf{z}) \geq 1 \quad \text{where } s(\mathbf{z}) : \mathbb{R}^2 \mapsto \mathbb{R} \text{ and } c(\mathbf{z}) : \mathbb{R}^2 \mapsto \mathbb{R} \text{ are monotone.} \quad (8)$$

We presume an upper and lower bound $(\mathbf{z}^u, \mathbf{z}^l)$ such that resulting search domain $\mathcal{Z} = [z_1^l, z_1^u] \times [z_2^l, z_2^u]$ contains the optimal solution $\mathbf{z}^* = \arg\min_{\mathbf{z}:c(\mathbf{z})\geq 1} s(\mathbf{z})$. Since both $s(\mathbf{z})$ and $c(\mathbf{z})$ are monotone we know that $\mathbf{z}^*$ must lie on or directly above the constraint boundary $c(\mathbf{z}) = 1$ (see Lemma 2).

In each iteration $k$ of Algorithm 1 we query a point $\mathbf{z}_k^q \in \mathcal{Z}$ and observe the corresponding objective and constraint values $s(\mathbf{z}_k^q)$ and $c(\mathbf{z}_k^q)$. We separate the queries into two sets $\mathcal{Q}^u$ and $\mathcal{Q}^l$ based on whether they lie above or below the constraint boundary. That is, we add $\mathbf{z}_k^q$ to $\mathcal{Q}^u$ if $c(\mathbf{z}_k^q) \geq 1$ and to $\mathcal{Q}^l$ otherwise. Since the optimal solution lies on the constraint boundary and $c(\mathbf{z})$ is monotone, we can rule out entire corners of the search domain: For each $\mathbf{z}^q \in \mathcal{Q}^u$ we can rule all points $\mathbf{z}' > \mathbf{z}_k^q$ as candidates for the optimal solution and, similarly for all $\mathbf{z}^q \in \mathcal{Q}^l$, we can rule out all $\mathbf{z}' \leq \mathbf{z}_k^q$. This also allows us to *prune* the sets $\mathcal{Q}^u$ and $\mathcal{Q}^l$ by removing all the points from them that can be ruled out by a new query results. To keep track which parts of $\mathcal{Z}$ have not been ruled out yet, we construct an *upper and lower frontiers*, here expressed as functions $z_1 \mapsto z_2$ and $z_2 \mapsto z_1$,

$$F_2^u(z_1; \mathcal{Q}^u) = \min\{z_2' \mid z_1 \geq z_1', \mathbf{z}' \in \mathcal{Q}^u\}, \quad F_1^u(z_2; \mathcal{Q}^u) = \min\{z_1' \mid z_2 \geq z_2', \mathbf{z}' \in \mathcal{Q}^u\} \quad (9)$$

$$F_2^l(z_1; \mathcal{Q}^l) = \max\{z_2' \mid z_1 \leq z_1', \mathbf{z}' \in \mathcal{Q}^l\}, \quad F_1^l(z_2; \mathcal{Q}^l) = \max\{z_1' \mid z_2 \leq z_2', \mathbf{z}' \in \mathcal{Q}^l\} \quad (10)$$

such that the points $\Gamma = \{(z_1, z_2) \in \mathcal{Z} \mid F_2^l(z_1; \mathcal{Q}^l) \leq z_2 \leq F_2^u(z_1; \mathcal{Q}^u)\} \subseteq \mathcal{Z}$ between the frontiers are still plausible candidates for the optimal solution. For notational brevity, we define $\mathcal{F}^u = \{\mathbf{z} \in \mathcal{Z} | F_1^u(z_2; \mathcal{Q}^u) = z_1 \lor F_2^u(z_1; \mathcal{Q}^u) = z_2\}$ and $\mathcal{F}^l$ analogously as the set of points that lie on the upper and lower frontier respectively.

At any point, the best solution to (8) that we can give, is the best query we have made so far that fulfills the constraint, i.e., $\hat{\mathbf{z}} = \arg\min_{\mathbf{z} \in \mathcal{Q}^u} s(\mathbf{z})$. If we assume Lipschitz continuity for $s$, which holds in our case holds since $\mathcal{Z}$ is bounded and the avg-std is differentiable in $l$ and $\nu$, we can bound how much our best solution is away from the optimum, i.e., $s(\mathbf{z}^*)$:

**Lemma 1.** *Let $s(\mathbf{z})$ be $L$ Lipschitz and $d(\Gamma, \mathcal{F}^u) := \max_{\mathbf{z}' \in \Gamma} \min_{\mathbf{z} \in \mathcal{F}^u} ||\mathbf{z} - \mathbf{z}'||$ the max-min distance between the frontiers. Then, the sub-optimality is bounded by $s(\hat{\mathbf{z}}) - s(\mathbf{z}^*) \leq L\, d(\Gamma, \mathcal{F}^u)$ .*

---

**Algorithm 2** Safe Meta-BO (SAMBO)

**Input:** Safe BO problem with $f^{tar}$ and $q^{tar}$, set of datasets $\{\mathcal{D}^f_{i,T_i}\}^n_{i=1}$, $\{\mathcal{D}^q_{i,T_i}\}^n_{i=1}$

1: **for** $h \in \{f, q\}$ **do**
2:    $(l_h, \nu_h) \leftarrow$ FRONTIERSEARCH$(\{\mathcal{D}^h_{i,T_i}\}^n_{i=1})$
3:    $\rho_{\boldsymbol{\theta}_h}(h) \leftarrow$ F-PACOH$(\{\mathcal{D}^h_{i,T_i}\}^n_{i=1}, \rho_{l_h, \nu_h}(h))$

   **Return:** $\hat{\mathbf{z}}^* \leftarrow$ SAFEBO$(f^{tar}, q^{tar}, \rho_{\boldsymbol{\theta}_f}(f), \rho_{\boldsymbol{\theta}_q}(q))$

---

Here, the key insight is that we can bound the sub-optimality $s(\hat{\mathbf{z}}) - s(\mathbf{z}^*)$ with the maximum distance of any point between the frontiers from its closest point on the upper frontier instead of $\hat{\mathbf{z}}$. This is the case because $\hat{\mathbf{z}}$ dominates any point on the upper frontier (i.e., $s(\hat{\mathbf{z}}) \leq s(\mathbf{z}') \; \forall \mathbf{z} \in \mathcal{F}^u$).

Hence, we want to choose the next query so that we can shrink the max-min distance $d(\Gamma, \mathcal{F}^u)$ between the frontiers the most. For this purpose, we select the largest max-min rectangle between the frontiers (see Definition 5). Then, we choose the query point, that reduce the max-min distance within the rectangle the best. For this we need to consider two scenarios that will affect the max-min distance differently: either the query point satisfies the constraint ($c(\mathbf{z}^q) \geq 1$) or it does not ($c(\mathbf{z}^q) < 1$). We compute the rectangle's max-min distance for both scenarios and choose the query-point that gives us the lowest max-min distance in the less-favorable (worst-case) scenario. For more details we refer to Appendix C. Finally, we provide worst-case rates for the proposed frontier search algorithm:

**Theorem 1.** *Under the assumptions of Lemma 1, Algorithm 1 needs no more than* $k \leq 3^{\lceil \log_2(1/\epsilon) \rceil} = \mathcal{O}\left((1/\epsilon)^{1.59}\right)$ *iterations to have a sub-optimality of less than* $s(\hat{\mathbf{z}}) - s(\mathbf{z}^*) \leq L \|\mathbf{z}^u - \mathbf{z}^l\| \, (1/\epsilon)$.

The $\mathcal{O}\left((1/\epsilon)^{1.59}\right)$ query complexity is more efficient than those of similar algorithms that do not make use of the monotonicity properties, e.g., that of grid search: $\mathcal{O}\left((1/\epsilon + 1)^2\right)$.

## 5   Safe BO with meta-learned GP priors

In Sec. 4 we discuss how to efficiently choose the kernel variance and lengthscale, so that we obtain calibrated and yet sharp uncertainty estimates. Now, we go one step further and use the related datasets $\mathcal{D}_{1,T_1}, ..., \mathcal{D}_{n,T_n}$ to meta-learn GP priors, aiming to give the downstream safe BO algorithm more prior knowledge about the optimization problem at hand so that it can be more query efficient.

For this, we use the F-PACOH [11] meta-learning approach, introduced in Sec 3. We choose this method since it regularizes the meta-learned model in the function space and, thus, is able to maintain reliability of the uncertainty estimates, even for out-of-distribution data. F-PACOH requires a Vanilla GP $\rho(h) = \mathcal{GP}(h(\mathbf{x})|0, k(\mathbf{x}, \mathbf{x}'))$ as a hyper-prior, towards which it regularizes the meta-learned GP $\rho_{\boldsymbol{\theta}}(h) = \mathcal{GP}(h(\mathbf{x})|m_{\boldsymbol{\theta}}(\mathbf{x}), k_{\boldsymbol{\theta}}(\mathbf{x}, \mathbf{x}'))$. Hence, we face a similar kernel hyperparameter choice problem as addressed in Section 4: If the kernel parameters of the hyper-prior are chosen such that the Vanilla GP itself yields over-confident uncertainty estimates, safe BO with the meta-learned prior will most likely turn out to be unsafe as well. Hence, we use the calibration-sharpness based frontier search to choose the kernel variance and lengthscale of the hyper-prior.

Since safe BO maintains separate models for the target $f(\mathbf{x})$ and constraint $q(\mathbf{x})$, we meta-learn a prior for each of the functions. We split the sets of data triplets $\mathcal{D}_{i,T_i} = \{(\mathbf{x}_{i,t}, \tilde{f}_{i,t}, \tilde{q}_{i,t})\}^{T_i}_{i=1}$ into separate datasets $\mathcal{D}^f_{i,T_i} = \{(\mathbf{x}_{i,t}, \tilde{f}_{i,t})\}^{T_i}_{t=1}$ and $\mathcal{D}^q_{i,T_i} = \{(\mathbf{x}_{i,t}, \tilde{q}_{i,t})\}^{T_i}_{t=1}$. First, we use frontier search to find kernel parameters $(l_f, \nu_f)$ and $(l_q, \nu_q)$ that give good sharpness subject to calibration on the respective set of datasets. Then, we meta-learn GP priors $\rho_{\boldsymbol{\theta}_f}(f)$ and $\rho_{\boldsymbol{\theta}_q}(q)$ for $f$ and $q$ with F-PACOH, while using the Vanilla GPs $\rho_{l_f, \nu_f}(f)$ and $\rho_{l_q, \nu_q}(q)$ with the chosen kernel hyper-parameters as hyper-prior. Finally, we run a safe BO algorithm (either SAFEOPT or GOOSE) with the meta-learned GP priors and perform safe Bayesian optimization on our target problem of interest. This procedure is summarized in Algorithm 2. We refer to our *Safe Meta-BO* (SaMBO) algorithm with GOOSE, as SAMBO-G and, when instantiated with SAFEOPT as SAMBO-S.

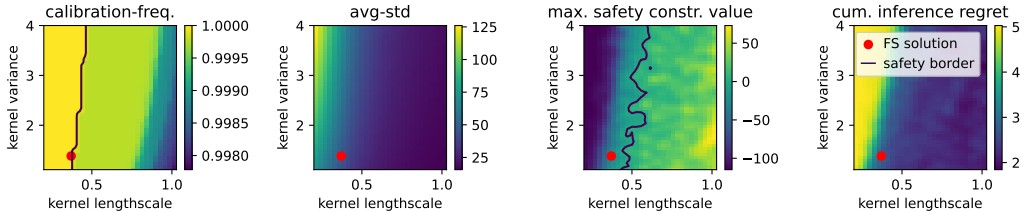

Figure 2: Left: calib and avg-std across a grid of kernel lengthscales $l_q$ and variances $\nu_q$ for the constraint model. Right: max safety constraint value and cumulative regret for 200 iterations of safe BO. Calibration and sharpness are good proxies for safety and efficiency of the downstream safe BO.

# 6 Experiments

First, we investigate if the calibration-/sharpness-based frontier search chooses good and safe kernel parameters. Second, we evaluate the the full SAMBO algorithm in a range of safe BO environments.

## 6.1 Experiment Setup

In the following, we describe our experiment setup and methodology. See Appx. F for more details.

**Synthetic safe BO benchmark environments.** We consider two synthetic environments, based on popular benchmark functions with a two-dimensional domain. The first is based on the *Camelback* function [54] overlaid with products of sinusoids of varying amplitude, phase and shifts. The constraint function is similar, with an additional random quadratic component to ensure connection between most of the safe regions of the domain. The second environment is based on the challenging Eggholder function [55] with many local minima. The function's shape can be varied by random sampling of three parameters. The constraint is a quadratic function, overlayed with sinusoids of varying frequencies. A task corresponds to a pair of randomly drawn target and constraint functions.

**Controller Tuning for a high-precision linear robot.** As robotic case study, we tune the controller of a linear axis in an Argus linear motion system by Schneeberger Linear Technology, a high-precision/speed robot for wafer inspection. To goal is to tune the three gain parameters of a cascaded PI controller to achieve minimal position error. We minimize the total variation (TV) of the position error signal, while constraining its maximum frequency in the FFT which measures (potentially damaging) instabilities/vibrations in the system. Different tasks correspond to different step sizes, ranging from $10\mu m$ to $10mm$. At different scales, the robot behaves differently in response to the controller parameters, resulting in different target and constraint functions. The experiments are conducted with a simulation of the robot, so we can explore unsafe kernel hyper-parameters – a key element for finding and visualizing the safety boundary in Figure 2.

**Meta-training data.** We generate the benchmark meta-training by running SAFEOPT with conservative kernel hyper-parameters choices on each task. The resulting datasets are non-i.i.d. and only consist of observations where $q(\mathbf{z}) < 0$, much more realistic than sampling data uniformly and i.i.d.. With this, we aim to mimic a practical scenario where we have performed various safe BO on related tasks in the past and now want to harness the corresponding data. For the synthetic environments, we use $n = 40$ tasks with $T_i = 100$ samples in case of the Camelback-Sin and $T_i = 200$ samples for the Random Eggholder environment. For the Argus controller tuning, we use $n = 20$ and $T_i = 400$.

**Evaluation metrics.** To evaluate the performance of various methods, we run safe BO with them on at least 4 unseen test tasks with each 5 seeds. To measure a method's query efficiency, we report the (safe) inference regret $r_t = f(\hat{\mathbf{x}}_t^*) - f(\mathbf{x}^*)$ where $\hat{\mathbf{x}}_t^* = \arg\min_{\mathbf{z} \in \mathcal{S}_t} \mu_t(\mathbf{x})$ is a method's best current guess for the safe minimizer of (1). In Fig. 2, we also report the cumulative regret $R_T := \sum_{t=1}^{T} r_t$.

## 6.2 Choosing the kernel parameters via calibration & sharpness based frontier search

We investigate how well the calibration and sharpness based frontier search (FS) for finding kernel hyper-parameters works in practice. For that, we consider the Argus controller tuning problem. Generally, we perform FS in the log-space of $l$ and $\nu$ since we found this to work better in practice.

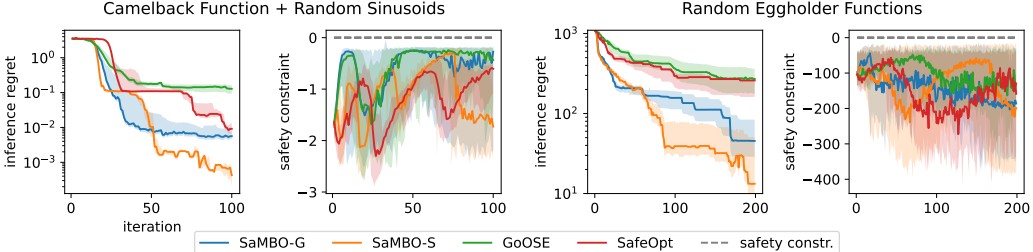

Figure 3: Safe BO regret and safety constraint on the synthetic benchmarks. SAMBO converges significantly faster towards the global optimum than the baselines, without violating the constraints.

Fig. 1 displays the frontier search at various iterations. FS quickly shrinks the set of possible safe optima (area between upper frontier (green) and lower frontier (red)) to points close to the safety border. After only 20 iterations, it already found a solution that is very close to the safe optimum. This showcases the efficiency we gain thanks to taking into account the monotonicity of the problem.

Furthermore, we investigate how well calibration and sharpness reflect what we care about: 1) No safety constraint violations and 2) query efficiency. We compute the maximum constraint value across 200 iterations with GOOSE as well as the cumulative regret across a grid of kernel length-scales and variances. Fig. 2 holds the results, together with the calib and avg-std. Overall, calibration and sharpness of the GPs' uncertainty estimates are a good proxy for the downstream safety during BO and, respectively, the regret. Importantly, all parameters that fulfill the calibration constraint $\text{calib}(\{\mathcal{D}_{i,T_i}\}_{i=1}^{n}, \omega) \geq 1$ lead to safe BO runs without constraint violations. Finally, the kernel parameters, chosen by FS, are both safe and lead to a small cumulative regret. Overall, this empirically supports the validity of our data-driven approach for choosing good, but safe, kernel parameters.

### 6.3 Safe Bayesian Optimization Benchmark & Controller Tuning

We compare the two SAMBO instantiations, SAMBO-S and SAMBO-G, with their corresponding safe BO baseline methods SAFEOPT and GOOSE. For the baselines, we use the GP kernel parameters found by FS. Fig. 3 displays the results for the synthetic benchmark functions and Fig. 4 for the Argus controller tuning. Note that, in case of the regret, the shaded areas correspond to confidence intervals while for the safety constraint values they correspond to the entire range of values (i.e., max - min). Overall,

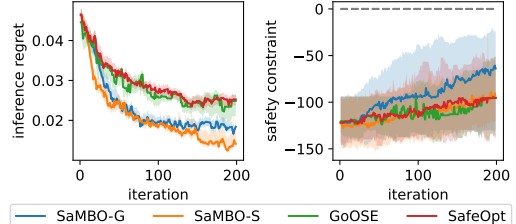

Figure 4: Safe controller tuning for the Argus robot. SAMBO safely finds good controller parameters faster than safe BO baselines.

SAMBO converges to near optimal solutions much faster than the baselines without meta-learning. Across all environments, there are no safety violations which demonstrates that 1) the kernel parameters by FS are safe and 2) SAMBO maintains safety thanks to principled regularization in the function space during meta-learning. The improved query efficiency without compromising safety in the controller tuning setting, where the constraint (maximum frequency in the signal) is highly non-smooth, demonstrates the applicability of SAMBO to challenging real-world robotics problems.

## 7 Summary and Discussion of Limitations

We have introduced a data-driven framework for choosing kernel parameters or even meta-learning GP priors that are both informative and reliable. When combined with a safe BO algorithm, the resulting SAMBO speeds up the optimization of, e.g., controller parameters, without compromising safety. Except for the observation noise variance, which is often known or easy to estimate, our framework makes safe BO free of hyper-parameters and, thus, more robust and practical. However, it relies on the availability of offline data that is both sufficient in quantity and representative of the target task. As our approach relies on empirical estimates of the calibration, it may fail to ensure safety when given too little data or tasks that are systematically different to the target task.

**Acknowledgments**

This research was supported by the European Research Council (ERC) under the European Union's Horizon 2020 research and innovation program grant agreement no. 815943, the Swiss National Science Foundation under NCCR Automation, grant agreement 51NF40 180545, and the Swiss Innovation Agency (Innosuisse), grant number 46716. Jonas Rothfuss was supported by an Apple Scholars in AI/ML fellowship. We thank Parnian Kassraie for proofreading the manuscript and making useful suggestions how to improve it.

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

# A  Safe Bayesian Optimization Algorithms

## A.1  SafeOpt

SAFEOPT [9, 16] make use of the GP confidence intervals, by defining a safe set on a discretized domain $\mathcal{X}$:

$$\mathcal{S}_t(\alpha) = \{\mathbf{x} \in \mathcal{X} \mid \mu_t^q(\mathbf{x}) + \beta(\alpha)\sigma_t^q(\mathbf{x}) < 0\}. \tag{11}$$

where we write $\mu_t^q(\mathbf{x})$ and $\sigma_t^q(\mathbf{x})$ short for $\mu^q(\mathbf{x}|\mathcal{D}_t)$ and $\sigma^q(\mathbf{x}|\mathcal{D}_t)$. Based on the safe set, SAFEOPT constructs two additional sets. First, the set of potential optimizers

$$\mathcal{M}_t(\alpha) = \{\mathbf{x} \in \mathcal{S}_t(\alpha) \mid \mu_t^f(\mathbf{x}) - \beta(\alpha)\sigma_t^f(\mathbf{x}) < \mu_t^f(\mathbf{x}^\dagger) + \beta(\alpha)\sigma_t^f(\mathbf{x}^\dagger)\}, \tag{12}$$

where $\mathbf{x}^\dagger$ is the best observed input so far. Second, the set of possible expanders

$$\mathcal{G}_t(\alpha) = \{\mathbf{x} \in \mathcal{S}_t(\alpha) \mid g(\mathbf{x}) > 0\}, \tag{13}$$

where $g_t(\mathbf{x})$ is defined as the additional number of inputs that become safe if we would query $\mathbf{x}$ and observe an hypothetical optimistic constraint value $\tilde{q} = \mu_t^q(\mathbf{x}') - \beta(\alpha)\sigma_t^q(\mathbf{x}')$.

$$g_t(\mathbf{x}) = |\{\mathbf{x} \in \mathcal{X} \setminus \mathcal{S}_t(\alpha) \mid \mu^q(\mathbf{x}'|\mathcal{D}_t \cup (\mathbf{x}, \tilde{q})) + \beta(\alpha)\sigma^q(\mathbf{x}'|\mathcal{D}_t \cup (\mathbf{x}, \tilde{q})) < 0\}|. \tag{14}$$

In each iteration of the optimization the sets are updated, w.r.t. the posterior belief of the underlying GPs a query candidate is selected from each the set of optimizers and the set of expanders. From the set of optimizers the GP-LCB sample

$$\mathbf{x}_{opt}^* = \arg\min_{\mathbf{z} \in \mathcal{M}_t(\alpha)} \mu_t^f(\mathbf{x}) - \beta(\alpha)\sigma_t^f(\mathbf{x}) \tag{15}$$

is selected. From the set of possible expanders

$$\mathbf{x}_{exp}^* = \arg\max_{\mathbf{x} \in \mathcal{G}_t(\alpha)} g_n(x) \tag{16}$$

is selected. Finally, SAFEOPT selects

$$\mathbf{x}^* = \arg\max_{\mathbf{x} \in \{\mathbf{x}_{opt}^*, \mathbf{x}_{exp}^*\}} \max\{\sigma_t^f(\mathbf{x}), \sigma_t^q(\mathbf{x})\} \tag{17}$$

as the next query.

## A.2  GoOSE

In SAFEOPT the expansion of the safe set is traded off against the optimization by the uncertainty of the prediction. This can lead to part-wise exploration of the safe set without consideration of the objective function. To cope with this [10] proposes a goal oriented safe exploration (GoOSE) algorithm. The idea is that the safe set is expanded only if it is also beneficial to the optimizaton of the objective. Like SAFEOPT it builds a GP model of the objective and constraints from noisy evaluations based on GP regression. The models of the constraints are used to construct two sets: First, the pessimistic safe set

$$\mathcal{S}_t^p(\alpha) = \{\mathbf{x} \in \mathcal{X} \mid \mu_t^q(\mathbf{x}) + \beta^q(\alpha)\sigma_t^q(\mathbf{x}) < 0\}, \tag{18}$$

which only contains domain points that are with high probability fulfilling the constraint. Second, we construct an optimistic safe set which includes all points we can optimistically expand the pessimistic safe set to. Rather than the version of Turchetta et al. [10] which requires an a-priori known Lipschitz constant, we use the version of König et al. [18] which only uses quantities that are provided by our GP model of the constraint function. For instance, the Lipschitz constant is approximated by the sup-norm $\|\nabla_{\mathbf{x}}\mu_t(\mathbf{x})\|_\infty$ of the gradient of the GP's posterior mean. This nicely accommodates non-stationary kernels which might are arise during meta-learning. We first make sure that we don't query the same point over and over by excluding all points within $\mathcal{S}_t^p(\alpha)$ whose confidence intervals for $q$ standard deviation become narrower than a alleatoric noise threshold $\epsilon$. The remaining points are:

$$W_t = \{x \in \mathcal{S}_t^p(\alpha) : 2\beta^q(\alpha)\sigma_t^q(x) > \epsilon\} \tag{19}$$

Then, we check for any $\mathbf{x} \in \mathcal{X}$ whether we still fulfill the safety constraint when we add the distance times the Lipschitz constant estimate to the optimistic lower bound $\mu_t^q(\mathbf{x}) - \beta^q(\alpha)\sigma_t^q(\mathbf{x})$ of the constraint CI, formally:

$$g_t(\mathbf{x}, \mathbf{z}) = \mathbb{I}\left[\mu_t^q(\mathbf{x}) - \beta^q(\alpha)\sigma_t^q(\mathbf{x}) + \|\nabla_{\mathbf{x}}\mu_t(\mathbf{x})\|_\infty \|\mathbf{x} - \mathbf{z}\|_2 < 0\right] . \tag{20}$$

Based on this indicator function, we can construct the optimistic safe set:

$$\mathcal{S}_t^o(\alpha) = \{\mathbf{x} \in \mathcal{X} \mid \exists \mathbf{z} \in W_t : g_t(\mathbf{x}, \mathbf{z}) = 1\} \tag{21}$$

Within the optimistic safe set $\mathcal{S}_t^o(\alpha)$, we now find the optimizer of a (standard) BO acquisition function. We use the the UCB aquisition function [42], i.e. $\text{aq}(\mathbf{x}) := \mu_t^f(\mathbf{x}) - \beta^f(\alpha)\sigma_t^f(\mathbf{x})$, to find the next query candidate $\tilde{\mathbf{x}}^* = \arg\min_{\mathbf{x} \in \mathcal{S}_t^o(\alpha)} \text{aq}(\mathbf{x})$.

If $\tilde{\mathbf{x}}^*$ is inside $\mathcal{S}_t^p(\alpha)$, it is evaluated. If not we query the point in $W_t$ which is closest to $\tilde{\mathbf{x}}^*$ and fulfills the expander criterion in (20). After querying a point and observing the corresponding $\tilde{f}_t$ and $\tilde{q}_t$, the posteriors of the GPs are updated and thus the sets which we defined above. This is repeated until $\tilde{\mathbf{x}}^*$ is inside $\mathcal{S}_t^p(\alpha)$ and can be evaluated or $\tilde{\mathbf{x}}^*$ is no longer in $\mathcal{S}_t^o(\alpha)$ and we compute a new query candidate.

The described procedure is summarized in Algorithm 3. Note that in comparison to Turchetta et al. [10], König et al. [18], Algorithm 3 not exclude points from $W_t$ that lie at the periphery of the domain because checking whether a point lies at the periphery is hard when working with uniformly sampled domain points instead of a grid.

---

**Algorithm 3** GoOSE algorithm

**Input:** Initial safe set $\mathcal{S}_0$
**Input:** GP models $f \sim \mathcal{GP}(\mu^f, k^f)$, $q \sim \mathcal{GP}(\mu^q, k^q)$
1: **for** $t = 1, ..., T$ **do**
2:      $\mathcal{S}_t^p(\alpha) \leftarrow \{\mathbf{x} \in \mathcal{X} \mid \mu_t^q(\mathbf{x}) + \beta(\alpha)\sigma_t^q(\mathbf{x}) < 0\}$          // pessimistic safe set
3:      $W_t \leftarrow \{x \in \mathcal{S}_t^p(\alpha) \mid 2\beta(\alpha)\sigma_t^q(x) > \epsilon\}$                // expanders
4:      $\mathcal{S}_t^o(\alpha) \leftarrow \{\mathbf{x} \in \mathcal{X} \mid \exists \mathbf{z} \in W_t : g_t(\mathbf{x}, \mathbf{z}) = 1\}$       // optimistic safe set
5:      $\tilde{\mathbf{x}}^* \leftarrow \arg\min_{\mathbf{x} \in \mathcal{S}_t^o(\alpha)} \text{aq}(\mathbf{x})$             // UCB candidate within optimistic safe set
6:      **if** $\tilde{\mathbf{x}}^* \in \mathcal{S}_t^p(\alpha)$ **then**
7:          evaluate $f(\tilde{\mathbf{x}}^*), q(\tilde{\mathbf{x}}^*),$
8:      **else**
9:          $\tilde{\mathbf{x}}_w \leftarrow \arg\min_{\mathbf{x} \in W_t} \|\mathbf{x} - \tilde{\mathbf{x}}^*\|_2$ s.t. $g_t(\mathbf{x}, \tilde{\mathbf{x}}^*) = 1$     // expand $\mathcal{S}$ in direction of $\tilde{\mathbf{x}}^*$
10:      evaluate $f(\tilde{\mathbf{x}}_w), q(\tilde{\mathbf{x}}_w),$
     **Return:** $\hat{\mathbf{x}}^* \leftarrow \arg\min_{\mathbf{x} \in \mathcal{S}_t^p(\alpha)} \mu_t^f(\mathbf{x})$       // return safe point with best posterior mean

---

# B   Meta-Learning reliable priors with F-PACOH

The F-PACOH method of Rothfuss et al. [11] uses a set of datasets $\mathcal{D}_{1,T_1}, ..., \mathcal{D}_{n,T_n}$ to meta-learn a GP prior. For that, it requires a parametric family $\{\rho_{\boldsymbol{\theta}} | \boldsymbol{\theta} \in \boldsymbol{\Theta}\}$ of GP priors $\rho_{\boldsymbol{\theta}}(h) = \text{GP}(h(\mathbf{x})|m_{\boldsymbol{\theta}}(\mathbf{x}), k_{\boldsymbol{\theta}}(\mathbf{x}, \mathbf{x}'))$. Typically the mean and kernel function of the GP are parameterized by neural networks. In addition, it presumes a Vanilla GP $\rho(h) = \text{GP}(h(\mathbf{x})|0, k(\mathbf{x}, \mathbf{x}'))$ as stochastic process hyper-prior. In our case, the hyper-prior GP has a zero-mean and a SE kernel. During the meta-training the marginal log-likelihood $\ln Z(\mathcal{D}_{i,T_i}, \rho_{\boldsymbol{\theta}}) = \ln p(\mathbf{y}_i^{\mathcal{D}} | \mathbf{X}_i^{\mathcal{D}}, \boldsymbol{\theta})$ is used to fit the $\rho_{\boldsymbol{\theta}}(h)$ to the meta-training data. Here we write $\mathcal{D}_{i,T_i} = (\mathbf{X}_i^{\mathcal{D}}, \mathbf{y}_i^{\mathcal{D}})$ for the matrix of function inputs and vector of targets of the respective dataset. Since we use GPs, the marginal log-likelihood can be computed in closed form as

$$\ln p(\mathbf{y}^{\mathcal{D}} | \mathbf{X}^{\mathcal{D}}, \boldsymbol{\theta}) = -\frac{1}{2}\left(\mathbf{y}^{\mathcal{D}} - \mathbf{m}_{\mathbf{X}^{\mathcal{D}}, \boldsymbol{\theta}}\right)^\top \tilde{\mathbf{K}}_{\mathbf{X}^{\mathcal{D}}, \boldsymbol{\theta}}^{-1} \left(\mathbf{y}^{\mathcal{D}} - \mathbf{m}_{\mathbf{X}^{\mathcal{D}}, \boldsymbol{\theta}}\right) - \frac{1}{2}\ln|\tilde{\mathbf{K}}_{\mathbf{X}^{\mathcal{D}}, \boldsymbol{\theta}}| - \frac{T}{2}\ln 2\pi \tag{22}$$

where $\tilde{\mathbf{K}}_{\mathbf{X}^{\mathcal{D}}, \boldsymbol{\theta}} = \mathbf{K}_{\mathbf{X}^{\mathcal{D}}, \boldsymbol{\theta}} + \sigma^2 I$, with kernel matrix $\mathbf{K}_{\mathbf{X}^{\mathcal{D}}, \boldsymbol{\theta}} = [k_{\boldsymbol{\theta}}(\mathbf{x}_l, \mathbf{x}_k)]_{l,k=1}^{T_i}$, likelihood variance $\sigma^2$, and mean vector $\mathbf{m}_{\mathbf{X}^{\mathcal{D}}, \boldsymbol{\theta}} = [m_{\boldsymbol{\theta}}(\mathbf{x}_1), ..., m_{\boldsymbol{\theta}}(\mathbf{x}_{T_i})]^\top$.

**Algorithm 4** F-PACOH [11]

---

**Input:** Datasets $\mathcal{D}_{1,T_1}, ..., \mathcal{D}_{n,T_n}$, parametric family $\{\rho_{\boldsymbol{\theta}} | \boldsymbol{\theta} \in \boldsymbol{\Theta}\}$ of priors, learning rate $\alpha$
**Input:** Stochastic process hyper-prior with marginals $\rho(\cdot)$

1: Initialize the parameters $\boldsymbol{\theta}$ of prior $\rho_{\boldsymbol{\theta}}$
2: **while** not converged **do**
3:     **for** $i = 1, ..., n$ **do**
4:        $\mathbf{X}_i = [\mathbf{X}_{i,s}^{\mathcal{D}}, \mathbf{X}_i^M]$, where $\mathbf{X}_{i,s}^{\mathcal{D}} \subseteq \mathbf{X}_i^{\mathcal{D}}, \mathbf{X}_i^M \stackrel{iid}{\sim} \mathcal{U}(\mathcal{X})$        // Sample measurement set
5:        Estimate or compute $\nabla_{\boldsymbol{\theta}} \ln Z(\mathbf{X}_i^{\mathcal{D}}, \rho_{\boldsymbol{\theta}})$ and $\nabla_{\boldsymbol{\theta}} KL[\rho_{\boldsymbol{\theta}}(\mathbf{h}^{\mathbf{X}_i}) || \rho(\mathbf{h}^{\mathbf{X}_i})]$
6:        $\nabla_{\boldsymbol{\theta}} J_{F,i} = -\frac{1}{T_i} \nabla_{\boldsymbol{\theta}} \ln Z(\mathcal{D}_{i,T_i}, \rho_{\boldsymbol{\theta}}) + \left( \frac{1}{\sqrt{n}} + \frac{1}{nT_i} \right) \nabla_{\boldsymbol{\theta}} KL[\rho_{\boldsymbol{\theta}}(\mathbf{h}^{\mathbf{X}_i}) || \rho(\mathbf{h}^{\mathbf{X}_i})]$
7:     $\boldsymbol{\theta} \leftarrow \boldsymbol{\theta} - \alpha \frac{1}{n} \sum_{i=1}^{n} \nabla_{\boldsymbol{\theta}} J_{F,i}$                // Update prior parameter

---

At the same time, during meta-training, $\rho_{\boldsymbol{\theta}}(h)$ is regularized towards the hyper-prior $\rho(h)$. We can only tractably assess the stochastic processes $\rho_{\boldsymbol{\theta}}(h)$ and $\rho(h)$ in a finite set of (measurement) points $\mathbf{X} := [\mathbf{x}_1, ..., \mathbf{x}_k] \in \mathcal{X}^k, k \in \mathbb{N}$ through their finite marginal distributions of function values $\rho(\mathbf{h}^{\mathbf{X}}) := \rho(h(\mathbf{x}_1), ..., h(\mathbf{x}_k))$ and $\rho_{\boldsymbol{\theta}}(\mathbf{h}^{\mathbf{X}}) := \rho_{\boldsymbol{\theta}}(h(\mathbf{x}_1), ..., h(\mathbf{x}_k))$ respectively. In particuar, for earch task, we construct measurement sets $\mathbf{X}_i = [\mathbf{X}_{i,s}^{\mathcal{D}}, \mathbf{X}_i^M]$ by selecting a random subset $\mathbf{X}_{i,s}^{\mathcal{D}}$ of the meta-training inputs $\mathbf{X}_i^{\mathcal{D}}$ as well as random points $\mathbf{X}_i^M \stackrel{iid}{\sim} \mathcal{U}(\mathcal{X})$ sampled independently and uniformly from the bounded domain $\mathcal{X}$.[2] In each iteration, we compute the KL-divergence between the marginal distributions of the stochastic processes in the sampled measurement sets. Since both stochastic processes are GPs, their finite marginals are Gaussians $\rho(\mathbf{h}_i^{\mathbf{X}}) = \mathcal{N}(\mathbf{0}, \mathbf{K}_{\mathbf{X}_i})$ and $\rho_{\boldsymbol{\theta}}(\mathbf{h}_i^{\mathbf{X}}) = \mathcal{N}(\mathbf{m}_{\mathbf{X}_i, \boldsymbol{\theta}}, \mathbf{K}_{\mathbf{X}_i, \boldsymbol{\theta}})$ and thus the KL-divergence available in close form. In expectation, over many iteration, we effectively minimize $\mathbb{E}_{\mathbf{X}_i}[KL[q(\mathbf{h}^{\mathbf{X}_i}) || \rho(\mathbf{h}^{\mathbf{X}_i})]]$.

Overall, the loss with the functional KL regularizer which we minimize in F-PACOH reads as:

$$\mathcal{L}(\boldsymbol{\theta}) = \frac{1}{n} \sum_{i=1}^{n} \left( -\frac{1}{T_i} \underbrace{\ln Z(\mathcal{D}_{i,T_i}, \rho_{\boldsymbol{\theta}})}_{\text{marginal log-likelihood}} + \left( \frac{1}{\sqrt{n}} + \frac{1}{nT_i} \right) \underbrace{\mathbb{E}_{\mathbf{X}_i}\left[ KL[\rho_{\boldsymbol{\theta}}(\mathbf{h}^{\mathbf{X}}) || \rho(\mathbf{h}^{\mathbf{X}})] \right]}_{\text{functional KL-divergence}} \right). \quad (23)$$

The stochastic minimization procedure for (23) which we have described is summarized in Algorithm 4.

## C   Frontier Search

We aim to solve the constraint optimization problem

$$\min_{\omega} \; s(\mathbf{z}) \quad \text{s.t.} \; c(\mathbf{z}) \geq 1 \quad (24)$$

where $s : \mathbb{R}^2 \mapsto \mathbb{R}$ and $c : \mathbb{R}^2 \mapsto \mathbb{R}$ are monotonically increasing functions. Formally, we define the monotonicity w.r.t. to the partial order on $\mathbb{R}^2$:

**Definition 1** (Monotone Function). *A function* $h(\mathbf{z}) : \mathbb{R}^2 \mapsto R$ *is said to be monotone if for all* $(z_1, z_2), (z_1', z_2') \in \mathbb{R}^2$

$$z_1 \leq z_1' \wedge z_2 \leq z_2' \Rightarrow h(z_1, z_2) \leq h(z_1', z_2') . \quad (25)$$

For brevity we write $\mathbf{z} \leq \mathbf{z}'$ short for $z_1 \leq z_1' \wedge z_2 \leq z_2'$ and $\mathbf{z} \geq \mathbf{z}' \Leftrightarrow z_1 \geq z_1' \wedge z_2 \geq z_2'$.

For our algorithm, we assume knowledge of an upper bound $\mathbf{z}^u$ and lower bound $\mathbf{z}^l$ of the optimal solution $\mathbf{z}^*$ such that the rectangle $\mathcal{Z} = [z_1^l, z_1^u] \times [z_2^l, z_2^u]$ that is spanned by the bounds and contains $\mathbf{z}^*$.

**Assumption 1** (Valid Search Domain). *The search domain* $\mathcal{Z} = [z_1^l, z_1^u] \times [z_2^l, z_2^u]$ *is valid if it contains the optimum* $\mathbf{z}^* = \arg\min_{\mathbf{z}:c(\mathbf{z}) \geq 1} s(\mathbf{z})$ *of the constraint optimization problem.*

---

[2]In Section 3.3, we have portrayed the measurement sets as only the uniform domain points $\mathbf{X}_i^M$ for brevity of the exposition. However, our implementation uses $\mathbf{X}_i = [\mathbf{X}_{i,s}^{\mathcal{D}}, \mathbf{X}_i^M]$, as described here in the appendix.

Due to the monotonicity of $s(\mathbf{z})$ and $c(\mathbf{z})$ we know that the optimal solution must lie on or immediately above the constraint boundary. The latter case is rather a technical detail which is due to the fact that we do not assume continuity of $c$. Formally we have:

**Lemma 2** ($\mathbf{z}^*$ lies on or directly above the constraint boundary). *Let $\mathbf{z}^* = \arg\min_{\mathbf{z}:c(\mathbf{z})\geq 1} s(\mathbf{z})$ be the unique minimizer, then there exists no $z_2 \in [z_2^l, z_2^u]$ with $1 \leq c(z_1^*, z_2) < c(\mathbf{z}^*)$.*

*Proof.* The proof of Lemma 2 follows by reduction: Assume $\exists z_2 \in [z_2^l, z_2^u]$ with $1 \leq c(z_1^*, z_2) < c(\mathbf{z}^*)$. Since $c$ is monotonically increasing $c(z_1^*, z_2) < c(\mathbf{z}^*)$ implies that $z_2 < z_2^*$. Thus, by the monotonicity of $s$, we have that $s(z_1^*, z_2) \leq s(\mathbf{z}^*)$ and $\mathbf{z}^*$ cannot be the unique constrained minimizer. $\qquad\square$

In each iteration $k$ of Algorithm 1 we query a point $\mathbf{z}_k^q \in \mathcal{Z}$ and observe the corresponding objective and constraint values $s(\mathbf{z}_k^q)$ and $c(\mathbf{z}_k^q)$. Due to Lemma 2 and the monotonicity of $q$ we can rule out an entire corner of the search domain. In particular, if $c(\mathbf{z}_k^q) \geq 0$, we can rule all points $\mathbf{z}' > \mathbf{z}_k^q$ as candidates for the optimal solution and, similarly, if $c(\mathbf{z}_k^q) < 0$, we can rule out all $\mathbf{z}' \leq \mathbf{z}_k^q$.

To keep track of all the areas of the search domain we can rule out, we construct an upper and a lower frontier such that all ruled-out points lie either above the upper and below the lower frontier. To construct these frontiers, we separate all queries based on whether they fulfill the constraint or not. Initially, we set $\mathcal{Q}^u = \{\mathbf{z}^u\}$ and $\mathcal{Q}^l = \{\mathbf{z}^l\}$, since Assumption 1 together with Lemma 2 imply that $c(\mathbf{z}^u) \geq 1$ and $c(\mathbf{z}^l) \leq 1$. Then, for every query, we add $\mathbf{z}_k^q$ to $\mathcal{Q}^u$ if $c(\mathbf{z}^q) \geq 1$ and to $\mathcal{Q}^l$ otherwise.

We define the upper and lower frontiers as maps $z_1 \mapsto z_2$ and $z_2 \mapsto z_1$:

$$F_2^u(z_1; \mathcal{Q}^u) = \min\{z_2' \mid z_1 \geq z_1', \mathbf{z}' \in \mathcal{Q}^u\}, \quad F_1^u(z_2; \mathcal{Q}^u) = \min\{z_1' \mid z_2 \geq z_2', \mathbf{z}' \in \mathcal{Q}^u\} \quad (26)$$

$$F_2^l(z_1; \mathcal{Q}^l) = \max\{z_2' \mid z_1 \leq z_1', \mathbf{z}' \in \mathcal{Q}^l\}, \quad F_1^l(z_2; \mathcal{Q}^l) = \max\{z_1' \mid z_2 \leq z_2', \mathbf{z}' \in \mathcal{Q}^l\} \quad (27)$$

For convenience, we define the sets of points that lie on the frontiers as

$$\mathcal{F}^u(\mathcal{Q}^u) = \{\mathbf{z} \in \mathcal{Z} | F_1^u(z_2; \mathcal{Q}^u) = z_1 \vee F_2^u(z_1; \mathcal{Q}^u) = z_2\}, \quad (28)$$

$$\mathcal{F}^l(\mathcal{Q}^l) = \{\mathbf{z} \in \mathcal{Z} | F_1^l(z_2; \mathcal{Q}^l) = z_1 \vee F_2^l(z_1; \mathcal{Q}^l) = z_2\}. \quad (29)$$

If we assume Lipschitz continuity for $s$, we can bound how much our best solution is away from the optimum, i.e., $s(\mathbf{z}^*)$:

**Lemma 3** (Long version of Lemma 1). *Let $\mathbf{z}^* = \arg\min_{\mathbf{z}:c(\mathbf{z})\geq 1} s(\mathbf{z})$ be the solution of the constraint optimization problem where $s : \mathbb{R}^2 \mapsto \mathbb{R}$ is monotone and $L$ Lipschitz, and $c : \mathbb{R}^2 \mapsto \mathbb{R}$ is monotone constraint. Let*

$$\Gamma(\mathcal{Q}^l, \mathcal{Q}^u) = \{(z_1, z_2) \in \mathcal{Z} \mid F^l(z_1; \mathcal{Q}^l) \leq z_2 \leq F^u(z_1; \mathcal{Q}^u)\} \quad (30)$$

*be the set of points that lie between the frontiers and $\hat{\mathbf{z}} = \arg\min_{\mathbf{z}^q \in \mathcal{Q}^l} s(\mathbf{z}^q)$ the current best solution. Then, we always have that*

$$s(\hat{\mathbf{z}}) - s(\mathbf{z}^*) \leq L \underbrace{\max_{\mathbf{z}' \in \Gamma} \min_{\mathbf{z} \in \mathcal{F}^u} ||\mathbf{z} - \mathbf{z}'||}_{d(\Gamma, \mathcal{F}^u)}. \quad (31)$$

*where $d(\Gamma, \mathcal{F}^u) := \max_{\mathbf{z}' \in \Gamma} \min_{\mathbf{z} \in \mathcal{F}^u} ||\mathbf{z} - \mathbf{z}'||$ is the max-min distance between the frontiers.*

*Proof.* By Assumption 1, we know that $\mathbf{z}^* \in \mathcal{Z}$. Due to Lemma 2, and the construction of the upper and lower Frontiers (i.e. $F^u(z_1) < z_2 \Rightarrow c(z_1, c_2) \geq 1$ and $F^u(z_1) > z_2 \Rightarrow c(z_1, c_2) \geq 1$) we always have that $F^l(z_1^*) \leq z_2^* \leq F^u(z_1^*)$, that is, $\mathbf{z}^* \in \Gamma$. Due to the monotonicity of $s$, we have $\forall \mathbf{z} \in F^u$ that

$$s(\hat{\mathbf{z}}) - s(\mathbf{z}^*) = \underbrace{(s(\hat{\mathbf{z}}) - s(\mathbf{z}))}_{\leq 0} - (s(\mathbf{z}^*) - s(\mathbf{z})) \quad (32)$$

$$\leq s(\mathbf{z}) - s(\mathbf{z}^*) \quad (33)$$

$$\leq L||\mathbf{z} - \mathbf{z}^*|| \quad (34)$$

where the last step follows from the Lipschitz property of $s$. Finally, as $\mathbf{z}^* \in \Gamma$ we have take the maximum over $\mathbf{z}' \in \Gamma$ so that the bound holds in the worst case. However, at the same time, we can take the minimum over $\mathbf{z} \in \mathcal{F}^u$ since (34) holds for all points $\mathbf{z}$ on the upper frontier. Both steps yield the final result

$$s(\hat{\mathbf{z}}) - s(\mathbf{z}^*) \le L \max_{\mathbf{z}' \in \Gamma} \min_{\mathbf{z} \in \mathcal{F}^u} ||\mathbf{z} - \mathbf{z}'|| \,, \tag{35}$$

which concludes the proof. $\qquad \square$

The key insight of Lemma 3 is that we can bound the sub-optimality $s(\hat{\mathbf{z}}) - s(\mathbf{z}^*)$ with the maximum distance of any point between the frontiers from its closest point on the upper frontier instead of $\hat{\mathbf{z}}$. This is the case because $\hat{\mathbf{z}}$ dominates any point on the upper frontier (i.e., $s(\hat{\mathbf{z}}) \le s(\mathbf{z}') \; \forall \mathbf{z} \in \mathcal{F}^u$). In fact, we can further reduce the set of points which we have to consider for computing the max-min distance $d(\Gamma, \mathcal{F}^u)$ to the outer corner points of $\Gamma$, as defined in the following:

**Definition 2** (Outer Corner Points of $\Gamma$). *Let* $\mathbf{b}^l = (F_1^l(z_2^u), z_2^u)$ *be the left upper and* $\mathbf{b}^r = (z_1^u, F_2^l(z_1^u))$ *the right outer corner points of* $\Gamma$. *Let* $\mathcal{Q}_{\cup \mathbf{b}}^l = \mathcal{Q}^l \cup \{\mathbf{b}^l, \mathbf{b}_r\}$ *and* $(\mathbf{z}_1, ..., \mathbf{z}_{|\mathcal{Q}^l|+2})$ *its ordering such that* $z_{1,i}, \le z_{1,i+1}$. *We define*

$$\Gamma_{out}^l(\mathcal{Q}^l) := \{(z_{k-1,1}, z_{k,2}) \mid i = 2, ..., |\mathcal{Q}^l| + 2\} \cup \{\mathbf{b}^l, \mathbf{b}_r\} \tag{36}$$

*as the the outer corner points of* $\Gamma$.

This allows to compute the max-min distance $d(\Gamma, \mathcal{F}^u)$ by maximizing over a finite set of outer corner points, instead of the whole inter-frontier area $\Gamma$:

**Lemma 4** (Version of the max-min distance that is easier to compute). *Let* $\Gamma_{out}^l$ *be the outer corner points of* $\Gamma$, *as defined in (40). Then, we have that*

$$d(\Gamma, \mathcal{F}^u) = \max_{\mathbf{z} \in \Gamma_{out}^l} \min_{\mathbf{z}' \in \mathcal{F}^u} ||\mathbf{z} - \mathbf{z}'|| \,. \tag{37}$$

*Proof.* Due the construction of the frontiers, for every tuple $(\mathbf{z}, \mathbf{z}') \in \Gamma \times \mathcal{F}^u$ that is the optimal solution of the max-min problem in (31) there must be a $(\tilde{\mathbf{z}}, \mathbf{z}') \in \Gamma_{out}^l \times \mathcal{F}^u$ that is the optimal solution of the max-min problem on the RHS of (37). Again, we can show this by reduction: Assume this is not the case, and $\mathbf{z} \in \Gamma \setminus \Gamma_{out}^l$, then, by the geometrical properties of $\Gamma$ there exists a $\tilde{\mathbf{z}} \in \Gamma_{out}^l$ which we can obtain by reducing $z_1$ or $z_2$ of $\mathbf{z} = (z_1, z_2)$, such that $||\tilde{\mathbf{z}} - \mathbf{z}'|| > ||\mathbf{z} - \mathbf{z}'||$. Thus $(\mathbf{z}, \mathbf{z}')$ cannot be the optimal solution of the max-min problem. $\qquad \square$

As we only have to consider a finite set of outer corners, instead of the whole $\Gamma$, this makes the max-min distance easier to implement in practice.

Since the max-min distance bounds how sub-optimal our current best solution can be, we want to choose the next query so that we can shrink the max-min distance $d(\Gamma, \mathcal{F}^u)$ between the frontiers. For this purpose, we find a rectangle

$$\mathcal{R}_{\mathbf{z}, \mathbf{z}'} = \{\tilde{\mathbf{z}} \in \mathcal{Z} \mid \min\{z_1, z_1'\} \le \tilde{z}_1 \le \max\{z_1, z_1'\} \wedge \min\{z_2, z_2'\} \le \tilde{z}_2 \le \max\{z_2, z_2'\}\} \tag{38}$$

within $\Gamma$ that has the largest max-min distance

$$d_{\mathcal{R}_{\mathbf{z}, \mathbf{z}'}}(\Gamma, \mathcal{F}^u) = d(\Gamma \cap \mathcal{R}_{\mathbf{z}, \mathbf{z}'}, \mathcal{F}^u \cap \mathcal{R}_{\mathbf{z}, \mathbf{z}'}) \,. \tag{39}$$

To make finding this rectangle tractable, we can narrow down the points on the upper frontier which we have to consider to its corner points, defined in the following:

**Definition 3** (Boundary Points of Upper Frontier). *Let* $\mathcal{Z} = [z_1^l, z_1^u] \times [z_2^l, z_2^u]$ *be the search domain. Then*

$$\mathbf{b}_v^u = (z_1^u, \min\{z_2 | (z_1, z_2) \in \mathcal{F}^u\}) \,, \quad \mathbf{b}_h^u = (\min\{z_1 | (z_1, z_2) \in \mathcal{F}^u\}, z_2^u)$$

*are the points of the upper frontier that intersect the vertical and horizontal domain boundary.*

**Definition 4** (Corner Points of $\mathcal{F}^u$). *Let $(\mathbf{z}_1, ..., \mathbf{z}_{|\mathcal{Q}^u|+2})$ the ordering of $\mathcal{Q}^u \cup \{\mathsf{b}_v^u, \mathsf{b}_h^u\}$ such that $z_{1,i}, \leq z_{1,i+1}$ and $z_{2,i} \geq z_{2,i+1}$. Then*

$$\mathcal{F}_{cor}^u(\mathcal{Q}^u) := \{(z_{k-1,1}, z_{k,2}) \mid i = 2, ..., |\mathcal{Q}^u| + 2\} \cup \mathcal{Q}^u \cup \{\mathbf{b}^l, \mathbf{b}_r\} \tag{40}$$

*are the corner points of the upper frontier $\mathcal{F}^u$.*

Now, we consider all outer corners of the inter frontier area $\Gamma_{\text{out}}^l(\mathcal{Q}^l)$ and corners of the upper frontier $\mathcal{F}_{cor}^u(\mathcal{Q}^u)$ to find the largest max-min rectangle:

**Definition 5** (Largest Max-Min Rectangle). *The largest max-min rectangle, defined by its corner points $(\mathbf{z}, \mathbf{z}') \in \Gamma_{out}^l \times \mathcal{F}_{cor}^u$ is the rectangle in $\Gamma$ with the largest max-min distance such that it fulfills condition (3), formally:*

$$\text{LARGESTMAXMINRECT}(\mathcal{Q}^l, \mathcal{Q}^u) = \underset{(\mathbf{z}, \mathbf{z}') \in \Gamma_{out}^l(\mathcal{Q}^l) \times \mathcal{F}_{cor}^u(\mathcal{Q}^u)}{\arg\max} d_{\mathcal{R}_{\mathbf{z}, \mathbf{z}'}}(\Gamma, \mathcal{F}^u) \; s.t. \; (1) \wedge (2) \wedge (3)$$

*(1) rectangle lies in $\Gamma$, i.e. $(z_1, z_2') \in \Gamma \wedge (z_1', z_2) \in \Gamma$*

*(2) rectangle has a positive area, i.e. $|z_1 - z_1'| \cdot |z_2 - z_2'| > 0$*

*(3) we cannot obtain a smaller rectangle $(\mathbf{z}, \tilde{\mathbf{z}})$ with a upper frontier evaluation $\tilde{\mathbf{z}} \in \mathcal{Q}^u$ that matches $(\mathbf{z}, \mathbf{z}')$ in one side of the rectangle but has a smaller second side, i.e.*

$$\neg \, \exists \, \tilde{\mathbf{z}} \in \mathcal{Q}^u : (z_1 < \tilde{z}_1 < z_1' \wedge z_2' = \tilde{z}_2) \vee (z_2 < \tilde{z}_2 < z_2' \wedge z_1' = \tilde{z}_1) \tag{41}$$

Given the largest max-min rectangle $\mathcal{R}_{\mathbf{z}, \mathbf{z}'}$, we want to choose the next query point so that we can reduce the max-min distance $d_{\mathcal{R}_{\mathbf{z}, \mathbf{z}'}}(\Gamma, \mathcal{F}^u)$ within this rectangle as efficiently as possible. We consider the set of query candidates

$$\mathcal{Q}_{\mathbf{z}, \mathbf{z}'} = \Big\{ \underbrace{(z_1/2 + z_1'/2, z_2/2 + z_2'/2)}_{\text{center of rect.}}, \underbrace{(z_1', z_2/2 + z_2'/2)}_{\text{middle of right side}}, \underbrace{(z_1/2 + z_1'/2, z_2')}_{\text{middle of upper side}} \Big\}, \tag{42}$$

consisting of the center point, together with the middle points of its right/upper sides. From this query set, we choose the candidate that minimizes the worst-case max-min distance. In particular, if we query a point $\mathbf{z}^q$ there are two possible scenarios that will affect the max-min distance differently: either the point satisfies the constraint $(c(\mathbf{z}^q) \leq 1)$ or it does not $(c(\mathbf{z}^q) < 1)$. We compute the rectangle's max-min distance for both scenarios and choose the query-point that gives us the lowest max-min distance in the less-favorable (worst-case) scenario:

$$\text{BESTWORSTCASEQUERY}(\mathbf{z}, \mathbf{z}', \mathcal{Q}^l, \mathcal{Q}^u) =$$

$$\underset{\mathbf{z}_q \in \mathcal{Q}_{\mathbf{z}, \mathbf{z}'} \setminus (\mathcal{F}^l \cup \mathcal{F}^u)}{\arg\min} \max \Big\{ d_{\mathcal{R}_{\mathbf{z}, \mathbf{z}'}} \left( \Gamma(\mathcal{Q}^l \cup \mathbf{z}^q, \mathcal{Q}^u), \mathcal{F}^u(\mathcal{Q}^u) \right), \tag{43}$$

$$d_{\mathcal{R}_{\mathbf{z}, \mathbf{z}'}} \left( \Gamma(\mathcal{Q}^l, \mathcal{Q}^u \cup \mathbf{z}^q), \mathcal{F}^u(\mathcal{Q}^u \cup \mathbf{z}^q) \right) \Big\} .$$

If one of the query candidates already lies on one of the frontiers, it cannot expand the frontiers and thus also not improve the worst-case distance. Hence, we directly exclude such query candidates by removing them from the argmin in (43).

**Theorem 2** (Appendix version of Theorem 1). *Under the assumptions of Lemma 3 the Algorithm, 1 needs no more than $k \leq 3^{\lceil \log_2(1/\epsilon) \rceil} = \mathcal{O}\left( (1/\epsilon)\rceil^{1.59} \right)$ iterations to get*

$$s(\hat{\mathbf{z}}_k) - s(\mathbf{z}^*) \leq L ||\mathbf{z}^u - \mathbf{z}^l|| \, (1/\epsilon) . \tag{44}$$

*close to the optimal solution.*

*Proof.* In the worst case, it requires Algorithm 1 no more than three queries to half the max-min distance within a max-min rectangle. The process of doing so maximally segments the inter frontier area $\Gamma$ within the rectangle into three new max-min rectangles. This can be checked by going though the possible cases of how a max-min rectangle is split up by Algorithm 1. When a max-min rectangle

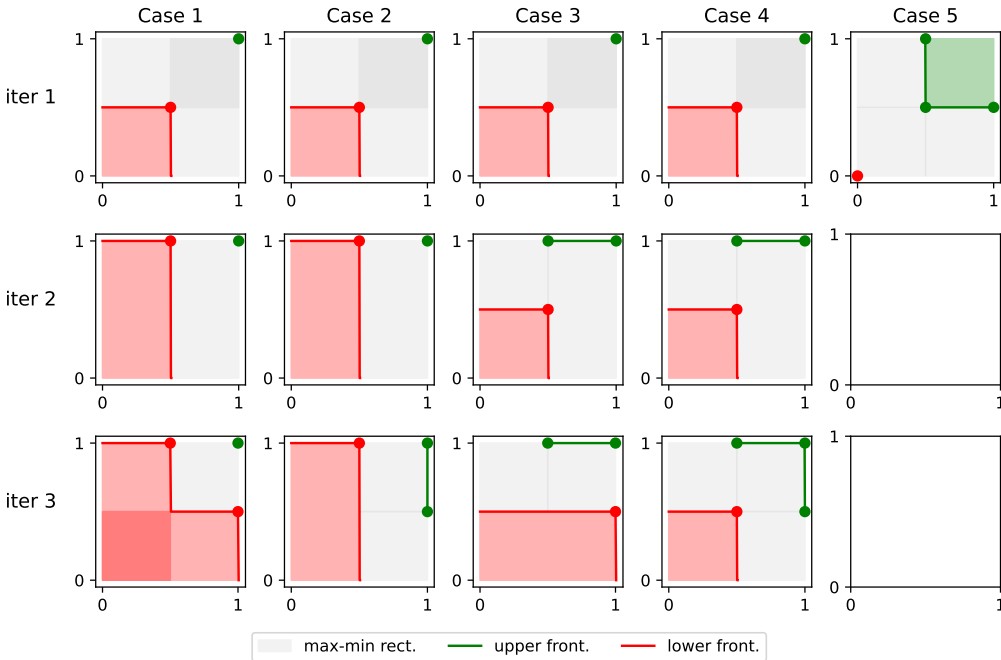

Figure 5: All possible cases how Algorithm 1 halfs the max-min distance within a max-min rectangle when only the upper right corner of the rectangle belongs to the upper frontier.

is split up in three iterations, there are, in principle, $8 = 2^3$ cases since each query can either lie above ore below the constraint boundary. However, when the first query fulfills the constraint, it immediately halfs the max-min distance. Thus no further queries are necessary and we effectively only have to consider 5 cases. Fig. 5 illustrates these five split-up cases for a max-min rectangle where only the upper right corner belongs to $\mathcal{F}^u$. Similarly, Fig. 6 and 7 illustrate the split-up cases when either half or the entire upper side of the rectangle belongs to $\mathcal{F}^u$. The cases when half or the entire right side of the rectangle belong to $\mathcal{F}^u$ are analogous. In all of these scenarios, only two further queries are necessary to half the max-min distance. Finally, in the case where both the upper and the right side of the rectangle belong half or full to $\mathcal{F}^u$ is trivial, as it only requires one more query. Thus, to shrink the max-min distance of all max-min rectangles by a factor of $n$, we need $k \le 3^{\lceil \log_2(n) \rceil}$ iterations at most. If we set $n = 1/\epsilon$ it follows by Lemma 3 that we need $k \le 3^{\lceil \log_2(1/\epsilon) \rceil}$ to obtain $d(\Gamma, \mathcal{F}^u) \le ||\mathbf{z}^u - \mathbf{z}^l||/\epsilon$. Finally, the result in the theorem follows from combining this with Lemma 3. □

## D    Implementation Details of SaMBO

**Calibration & Sharpness based Frontier Search.** To compute the calibration (5) and sharpness (6) across multiple datasets efficiently, we distribute the computation across individual processes for each datasets, using the ray distributed computing framework [56]. A model that always gives valid confidence intervals should always be calibrated, no matter what is the ordering of the datasets $\mathcal{D}_i$ which we split into train sets $\mathcal{D}_{i, \le t}$ and test sets $\mathcal{D}_{i, > t}$ to compute (5). Based on this principle, we compute avg-std and calib based on the given ordering as well as the reversed ordering of each dataset, and average both results. In principle, more permutations can be done, but come with additional computational cost.

We perform frontier search to search for the kernel lengthscale $l$ and variance $\nu$ parameter for both the target and constraint function model. For the constraint function model, we use the safety constraint $\text{calib}(\{\mathcal{D}^q_{i, T_i}\}^n_{i=1}, l_q, \nu_q) \ge 1$ (i.e. calib $= 1$ since calib $\in [0, 1]$. For the

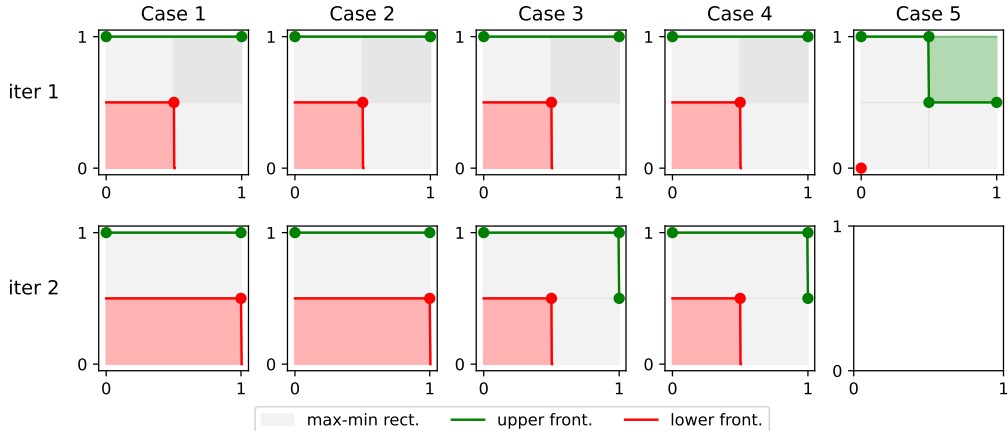

Figure 6: All possible cases how Algorithm 1 halfs the max-min distance within a max-min rectangle when the upper side of the rectangle belongs to the upper frontier.

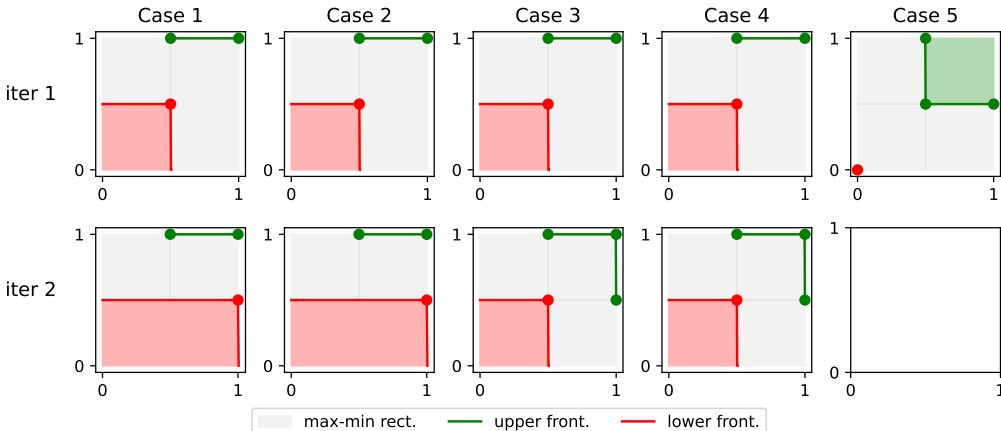

Figure 7: All possible cases how Algorithm 1 halfs the max-min distance within a max-min rectangle when half of the upper side of the rectangle belongs to the upper frontier.

target function model, the calibration requirements are not safety related and thus not as strict as in case of the constraint function. Hence, we only use a calibration frequency constraint $\mathrm{calib}(\{\mathcal{D}_{i,T_i}^f\}_{i=1}^n, l_f, \nu_f) \geq 0.95$ of 95 % when performing frontier search on $l_f$ and $\nu_f$. For both models, we run frontier search for 20 iterations as this gives us already a good solution without incurring to much computational cost. For the variances, we us an lower and upper boundary of 1.0 and 6.0 whereas for the lengthscale we consider the range [0.01, 5.0]. The frontier search is performed in the log-space to the base 10 of both variance and lengthscale, i.e., we transform the both variance and lengthscale with $t(z) = \log_{1} 0(z)$ during the frontier search and in the end transform back the result by $t^{-1}(z) = 10^z$.

**Parallelizing the sharpness and calibration computations.** Computing the calibration and sharpness metrics in (5) and (6) is computationally expensive since it requires computing the calib-freq and avg-std over all $n$ tasks and $T_i - 1$ subsequences within each task. Fortunately, each of these computations can be done independently. Thus, we use compute each of the summands in (5) and (6) in parallel processes and aggregate the results in the end. This significantly reduces the computation time for the calibration and sharpness which allows us to perform 20 steps of frontier search relatively fast.

**The NN-based GP prior.** Following [49, 11], we parameterize the GP prior $\rho_\theta(h) = \mathcal{GP}(h|m_\theta(\mathbf{x}), k_\theta(\mathbf{x}, \mathbf{x}'))$, particularly the mean $m_\theta$ and kernel function $k_\theta$, as neural networks (NN). Here, the parameter vector $\theta$ corresponds to the weights and biases of the NN. To ensure the positive-definiteness of the kernel, we use the neural network as feature map $\Phi_\theta(\mathbf{x}) : \mathcal{X} \mapsto \mathbb{R}^d$ that maps to a d-dimensional real-values feature space in which we apply a squared exponential kernel. We choose the dimensionality of the feature space the same as the domain. Accordingly, the parametric kernel reads as

$$k_\theta(x, x') = \nu_P \exp\left(-||\Phi_\theta(\mathbf{x}) - \Phi_\theta(\mathbf{x}')||^2/(2l_P)\right) \ . \tag{45}$$

Both $m_\theta(\mathbf{x})$ and $\Phi_\theta(\mathbf{x})$ are fully-connected neural networks with 3 layers with each 32 neurons and $\tanh$ non-linearities. The kernel variance $\nu_P$ and lengthscale $l_P$ are also learnable parameters which are appended to the NN parameters $\theta$. Since $l_P$ and $\sigma_P^2$ need to be positive, we represent and optimize them in log-space. Unlike Rothfuss et al. [11], we set the variance $\sigma^2$ of the Gaussian likelihood $p(y|h(\mathbf{x})) = \mathcal{N}(y; h(\mathbf{x}), \sigma^2)$ as a fixed parameter rather than meta-learning it. This is, because, in order to ensure safety, the likelihood variance needs to be the same as the one that was used during frontier search, when computing then calibration and sharpness.

**The hyper-prior.** We use a Vanilla GP $\mathcal{GP}(0, k(x, x'))$ as hyper-prior stochastic process. In that,

$$k(x, x') = \nu \exp\left(-||x - x'||^2/(2l)\right) \tag{46}$$

is a SE kernel with variance $\nu$ and lengthscale $l$ chosen by the frontier search procedure, discussed in Sec. 4.2.

**Minimization of the F-PACOH objective.** To estimate the F-PACOH objective in (23), we use measurement sets of size $k = 20$, i.e., considering a subset of 10 training points and 10 uniformly sampled domain points per iteration. We minimize the loss by performing 5000 iterations with the Adam optimizer with a learning rate of 0.001.

**Code and Data.** We provide implementations of the SaMBO components as well as the experiment scripts and data under `https://tinyurl.com/safe-meta-bo`.

# E  Further discussions about SaMBO

**Discussion on the applicability to higher-dimensional problems.** Similar to other BO / safe BO methods that are based on GPs, the sample complexity grows exponentially with the number of dimensions. The meta-learning in our case alleviates this issue to some degree by making the GP prior more informed about our environment of tasks in areas where meta-training data is available. However, to maintain safety, we regularize the meta-learned GP prior towards a Vanilla GP with SE kernel (cf. Eq. 3) in areas without or little meta-training data. The higher-dimensional our safe BO problem, the sparser is the meta-training data in the domain. Thus, the areas where our meta-learned GP resembles a Vanilla GP become larger in proportion and we face again the general issue of poor sample complexity in high dimensions. Generally, without additional assumption that, e.g., there exists some lower-dimensional sub-space in which the functions we are optimizing lie, we do not believe that the 'curse of dimensionality' problem can be solved while, at the same time, assuring safety. Generally, we not aware of any safe BO method that has been successfully applied to a high-dimensional optimization problem with safety constraint.

**Discussion on the monotonicity of the calibration-sharpness constraint optimization problem.** As discussed in Section 4.2, the introduced frontier search algorithm aims to exploit the monotonicity properties of the calibration-sharpness constraint optimization problem in 7. For large ranges of $l$ and $\nu$, the avg-std and avg-calib are monotonically increasing in the kernel variance $\nu$ and decreasing in the lengthscale $l$. While the monotonicity for the avg-std provably holds across the spectrum, the monotonicity of the calibration frequency in $l$ is only an empirical heuristic that holds in almost all cases if $\nu$ is at least as big as the variance of the targets $y$ in a dataset. If the kernel variance is chosen smaller than the variance of the data, significant parts of the function

that underlie the data are outside of the high-probability regions of the GP prior and the GPs predictive distribution systematically underestimates the corresponding variance in the absence of close-by data points. In such a cases, a larger lengthscale can actually improve the calibration by increasing the sphere of local influence of data points on the predictive distribution and, thus, slowing down the reversion of the posterior towards the prior with too small variance.

Figure 8 displays the calibration frequency in response to $l$ and $\nu$ in the same setting as Figure 2, but with an extended range where the kernel variance $\nu$ becomes smaller than 1 (data is standardized) and thus smaller than the variance of the data. The lower left corner where $\nu < 1$ illustrates the described breakdown of monotonicity.

In practice, this is not an issue since we can easily we standardize our data and then choose the lower bound of kernel variance in the frontier search to be 1. In the resulting search space, our constraint optimization problem is monotone and the frontier search converges as expected.

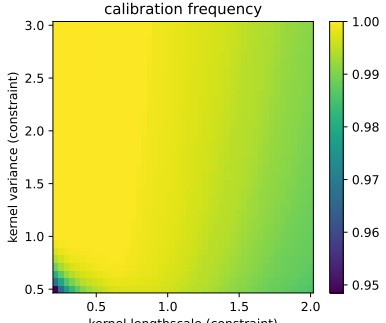

Figure 8: Calibration frequency for varying kernel lengthscales $l$ and variances $\nu$. The data underlying the calibration computations has been standardized. While avg-calib is monotone in $l$ for $\nu > 1$, the monotonicity breaks down if the kernel variance is smaller than the variance of the data.

## F   Experiment Details

### F.1   SafeBO environments

#### F.1.1   The Camelback + Random Sinusoids environment

The environment objective/target function corresponds to a Camelback function [54]

$$g(x_1, x_2) = \max\left(-(4 - 2.1 \cdot x_1^2 + x_1^4/3) * x_1^2 - x_1 x_2 - (4 \cdot x_2^2 - 4) * x_2^2, \ -2.5\right) . \quad (47)$$

plus random sinusoid functions, defined over the 2-dimensional cube $\mathcal{X} = [-2, 2] \times [-1, 2]$ as domain. Specifically, the target function is defined as

$$f(x_1, x_2) = g(x_1, x_2) + a \sin(\omega_f * (x_1 - \rho)) \sin(\omega_f * (x_2 - \rho)) \quad (48)$$

wherein the parameters are sampled independently as

$$a \sim \mathcal{U}(0.3, 0.5), \quad \omega_f \sim \mathcal{U}(0.2, 2.0), \quad \rho \sim \mathcal{N}(0, 1.0) . \quad (49)$$

The constraint function $q(\mathbf{x})$ is a linear combination of the camelback function $g(\mathbf{x})$, a product of sinusoids along the two dimensions and a quadratic component that ensures that the safe regions are sufficiently connected so that they can be reached:

$$q(x_1, x2) = 3 \cdot \sin(0.4 \cdot \pi \cdot \omega_q - 2) \cdot \sin(2\pi \cdot \omega_q) - b \cdot (x_1^2 + x_2^2) + 1.2 \cdot g(x_1, x_2) - 0.7 . \quad (50)$$

Here the parameters $\omega_q$ and $b$ are sampled as follows:

$$\omega_q \sim \mathcal{U}(0.45, 0.5), \quad b \sim \mathcal{U}(0.3, 0.5) \quad (51)$$

The optimization domain is $\mathcal{X} = [-2, 2] \times [-1, 1]$. As initial safe point we use $\mathcal{S}_0 = \{(-1.5, -0.5)\}$. Figure 9 displays an example task of the Camelback + Random Sinusoids environment.

#### F.1.2   The Random Eggholder environment

The random Eggolder environemnt is based on the Eggholder function [55], a popular benchmark function used in global optimization:

$$f(x_1, x_2) = -(x_2 + c) \cdot \sin\left(\sqrt{|ax_2 + x_1/2 + 47|}\right) - b \cdot x_1 \cdot \sin\left(\sqrt{|x_1 - x_2 - 47|}\right) \quad (52)$$

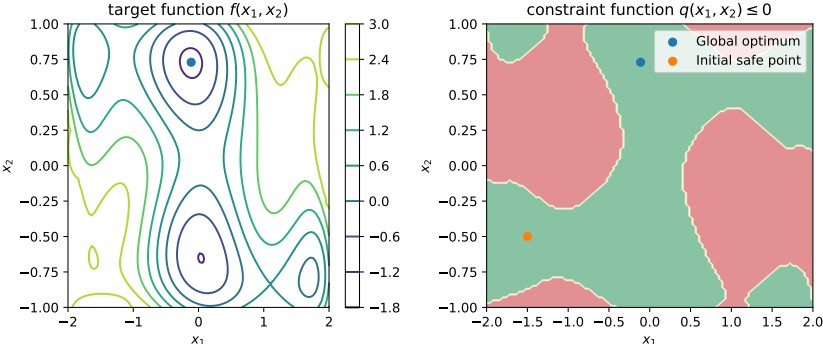

Figure 9: Example task of the Camelback + Random Sinusoids environment. Left: target function. Right: Safe regions in green and unsafe regions in red.

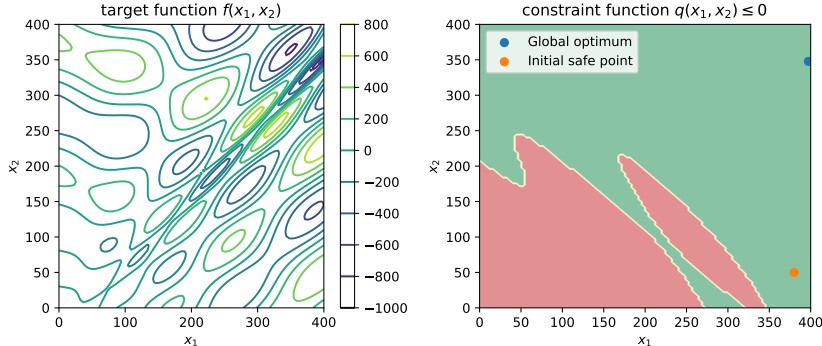

Figure 10: Example task of the Random Eggholder environment. Left: target function. Right: Safe regions in green and unsafe regions in red.

We randomly sample its parameters $a, b, c$ as follows:

$$a \sim \mathcal{U}(0.6, 1.4), \quad b \sim \mathcal{U}(0.6, 1.4), \quad c \sim \mathcal{N}(47, 5^2) \tag{53}$$

to obtain different tasks. The corresponding constraint function is defined as

$$q(x_1, x_2) = 300 - \sqrt{x_1^2 + 2x_2^2} + 50 \sin\left((\omega_1 x_1 + \omega_2 x_2)/20\right) , \tag{54}$$

where the frequencies are sampled independently as $\omega_1, \omega_2 \sim \mathcal{U}(0.8, 1.2)$. The optimization domain is $\mathcal{X} = [0, 400] \times [0, 400]$. As initial safe point we use $\mathcal{S}_0 = \{(380, 50)\}$. Figure 10 displays an example task of the Random Eggholder environment. The environment is particularly challenging since the Eggholder function has many local minima.

### F.1.3 Tuning controller parameters for the Argus linear robotic platform

The high-precision motion system Argus from Schneeberger Linear Technology is shown on Fig. 11). It is a 3 axes positioning system with 2 orthogonal linear axes and a rotational axes on top. The system has an accuracy of $\pm 10\mu$m, bidirectional repeatability of $0.7\mu$m and $3\sigma$ position stability of $< 1$nm. In our experiments we focus on the upper linear axis of the system.

The exact optimization problem is

$$\mathbf{x}^* = \underset{\mathbf{x}=[\text{PKP,VKP,VKI}]}{\arg\min} \quad T_s \cdot \frac{d}{dt}\text{pe}[t_{\text{move}} : t_{\text{end}}] \tag{55}$$

$$\text{s.t.} \quad \text{FFT}_{\max}(\mathbf{x}^*) = \max_{\omega \in [0.03, 0.07]} |\text{fft}(\text{ve})|(\omega) + b \cdot \max_{\omega \in [0.08, 0.1]} |\text{fft}(\text{ve})|(\omega) - \kappa < 0 \tag{56}$$

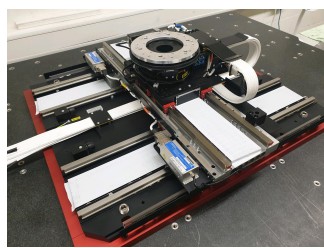

Figure 11: Argus motion system

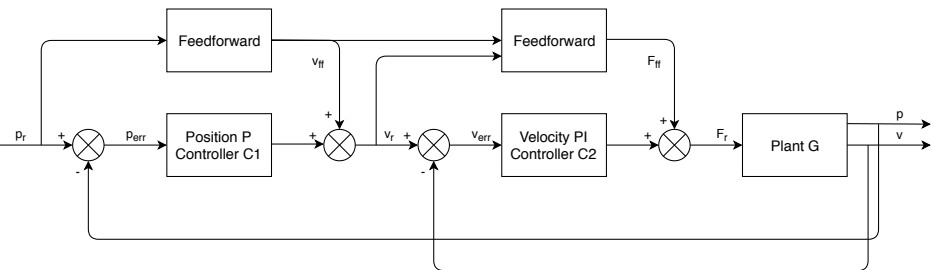

Figure 12: Argus controller structure

where PKP is the proportional gain of the position control loop, VKP is the proportional gain of the velocity control loop, VKI is the integral gain of the velocity control loop, $T_s$ is the settling time, pe is the position error measurement, ve is the velocity error measurement, $t_{\text{move}}$ is the move-time defined as the timepoint where the reference movement is finished, $t_{\text{end}}$ is the time endpoint of the movement measurement, set to 1.2s, $\omega$ is the frequency, fft is the fast Fourier transform in a given frequency window, $b = 5$ is a scaling factor, and $\kappa$ is the constraint limit, dependent on the reference stepsize. The position reference trajectory is a step-wise constant jerk based s-curve with jerk $= 200\frac{\text{m}}{\text{s}^3}$, maximum acceleration $a = 20\frac{\text{m}}{\text{s}^2}$ and maximum velocity $v = 1\frac{\text{m}}{\text{s}}$.

The simulation model of the Argus used in section 6.3 contains a cascade controller and a model of the Argus system (plant). The cascaded controller (see figure 12) was rebuilt from the real controller design and the plant (see figure 13) was modelled by 1) a fitted linear transfer function $G(s)$, containing a double integrator, the five most dominant resonances of form $T(s) = \frac{s^2/\omega_{ni}+2\lambda_{ni}/\omega_{ni}+1}{s^2/\omega_{di}+2\lambda_{di}/\omega_{di}+1}$ and a dead time, using frequency domain data and 2) a nonlinear position-dependent cogging model $F_c(p)$, based on linear interpolation of a lookup table for 16000 points at positions between $\pm200$mm of the axis using a discretization of $0.025$mm, where $p$ is position, $v$ is velocity, and $F_r$ is the force reference (proportional to current) applied to the motor of the axis. The linear transfer function has order 13. The resonant and anti-resonant frequencies are displayed in table 1, with $f_{n/d} = \omega_{n/d} \cdot \sqrt{1 - 2\lambda_{n/d}^2}$ and dead time $t_{\text{dead}} = 2$ms

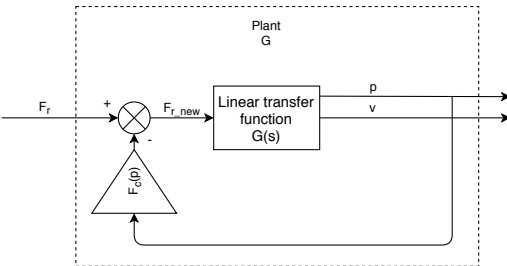

Figure 13: Argus controller structure

|   | $f_n$ in Hz | $\lambda_n$ | $f_d$ in Hz | $\lambda_d$ |
|---|---|---|---|---|
| 1 | 390 | 0.1 | 400 | 0.1 |
| 2 | 475 | 0.03 | 500 | 0.05 |
| 3 | 690 | 0.03 | 800 | 0.06 |
| 4 | 870 | 0.03 | 900 | 0.04 |
| 5 | 1050 | 0.03 | 1100 | 0.06 |

Table 1: Argus simulation linear transfer function parameters

| Environment | # meta-train tasks | # points per task | lengthscale constr model |
|---|---|---|---|
| Camelback + Random Sin | 40 | 100 | 0.5 |
| Random Eggolder | 40 | 200 | 0.4 |
| Argus Controller Tuning | 20 | 400 | 0.4 |

Table 2: Specifications of the meta-train data

The optimization domain is 3-dimensional and corresponds to the three controller gains PKP, VKP, VKI, restricted to the following ranges: $\mathcal{X} = [100, 400] \times [300, 1200] \times [500, 4000]$. As initial safe set, we take $\mathcal{S}_0 = \{(200, 800, 1000)\}$.

### F.2 Environment Normalization and Data Collection

**Environment Normalization.** The three environments, specified in Appx F.1 have vastly different scales, both in $\mathbf{x}$, $\tilde{f}$ and $\tilde{q}$. To alleviate the problems arising from different value ranges we *standardize* the environment data such that the value ranges are roughly those of a standard normally distributed random variable, before we pass on the data to the GP model. Hence, the kernel variances and lengthscales displayed in Fig. 6.2 correspond to the standardized value ranges.

To determine the standardization statistics (i.e., mean and standard deviation (std)) for $\mathbf{x}$ we use the domain ranges ($[x_i^l, x_i^u]$ for dimension $i = 1, ..., d$) of the environments. Assuming a uniform distribution over the domain, we use $\mu_{x_i} = \frac{x_i^u + x_i^l}{2}$ as mean and $\sigma_{x_i} = \sqrt{\frac{(x_i^u - x_i^l)^2}{12}}$. For $\tilde{f}$ and $\tilde{q}$ we are more conservative, because we do not want to underestimate their std which could lead to getting stuck in local optima or safety violations. In particular, we consider the respective minimum and maximum values of $\tilde{f}$ and $\tilde{q}$ observed in the union of the meta-train datasets, i.e. $\tilde{f}^{\min}$, $\tilde{f}^{\max}$, $\tilde{q}^{\min}$, $\tilde{q}^{\max}$. For $\tilde{f}$, we use $\mu_{\tilde{f}} = \frac{\tilde{f}^{\max} + \tilde{f}^{\min}}{2}$ and $\sigma_{\tilde{f}} = \frac{\tilde{f}^{\max} - \tilde{f}^{\min}}{3}$. For the constraint values, we set $\mu_{\tilde{q}} = 0$ because we do not want to distort the safety threshold and use $\sigma_{\tilde{q}} = \frac{\max\{|\tilde{q}^{\max}|, |\tilde{q}^{\min}|\}}{2}$ as std to re-scale the constraint values.

**Collection of meta-train data.** We collect the meta-training data by running SAFEOPT on each task. Table 2 holds the specifications for each of the three environments, i.e. the number of tasks $n$, the number of points/iterations $T_i$ per task and the constraint model lengthscale that is used for SAFEOPT. In all cases, we use a conservative lengthscale of 0.2 for the target model and a likelihood std of 0.1. Conservative parameter choices like these ensure that we continue exploring (within the confines of SAFEOPT) throughout the data collection and don not start to over-exploit, e.g. by querying the same data point many times.

### F.3 Parameter Choices for the Experiments

**Domain discretization.** Since both SAFEOPT and GOOSE require finite domains, we discretize the continuous domains of the safe BO tasks with 40000 uniform points each.

**Likelihood Std.** The standard deviation of the Gaussian likelihood of the GP is the only hyper-parameter we have to choose when employing our proposed frontier search in combination with Vanilla GPs. However, the observation noise is often known or easy to estimate by querying the

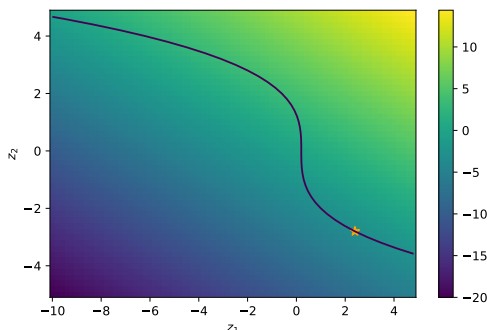

Figure 14: Monotone optimization problem in Eq. 57

same point multiple times and observing the variation of responses. We use the latter approach to determine the likelihood std in our experiments, obtaining the following settings: $\sigma = 0.02$ for the Camelback + Random Sin environment, $\sigma = 0.05$ for the Random Eggholder environment and $\sigma = 0.1$ for the Argus Controller parameter tuning.

**GoOSE expander sets.** For computing the GoOSE expander sets $W_t$, we always use $\epsilon = 0.2$. Note, that this applies after the environment standardization.

## G    Further Experiment Results

### G.1    Frontier Search

In this section, we aim to investigate how fast the proposed Frontier Search algorithm (c.f. Algorithm 1) approaches the optimal solution of a montone optimization problem like the one in (24). To this end, we use the following monotone, constraint optimization problem to test the algorithm:

$$\min_{\omega} s(\mathbf{z}) \quad \text{s.t.} \ \ c(\mathbf{z}) \geq 1$$
$$\text{with } s(\mathbf{z}) := z_1 + 2z_2 \tag{57}$$
$$q(\mathbf{z}) := 5 * z_1 + 0.5 * z_2^3 - 3$$

Figure 14 visualizes the $s(\mathbf{z})$ and the constraint boundary $q(\mathbf{z}) = 0$. In Figure 15 we visualize the first 33 iterations of Frontier Search on the optimization problem in (57). As we can see, Algorithm 1 quickly shrinks the area between the frontiers and returns solutions close to the optimum after only a handful of queries.

Finally, Figure 16 visualizes the convergence of Frontier Search, both in terms of the max-min distance and the sub-optimality $s(\hat{\mathbf{z}}^*) - s(\mathbf{z}^*)$ of the solution, together with the bounds provided in Theorem 1.

### G.2    Compute Time Analysis

Here, we provide a computational analysis of various components of the proposed SaMBO algorithm. Table 3 reports the time it takes to finish the computations associated with the frontier search as well as F-PACOH for different sizes of the meta-training data. Each experiment has been run on 16 processor cores of an Intel Xeon 8360Y 2.4GHz CPU. As in all the experiments, we use 20 iterations of frontier search and 5000 iterations of stochastic gradient descent (with the Adam optimized) on the F-PACOH objective.

As already discussed in Appendix D, we can use parallelization for the computation of the avg-calib and avg-std metric which makes frontier search relatively fast. For instance, for $n = 20$ tasks and $T = 200$ samples, the frontier search part of Algorithm 2 finishes in ca. 35 seconds. The computational complexity of the calibration and sharpness computations grows with $T$ since the number

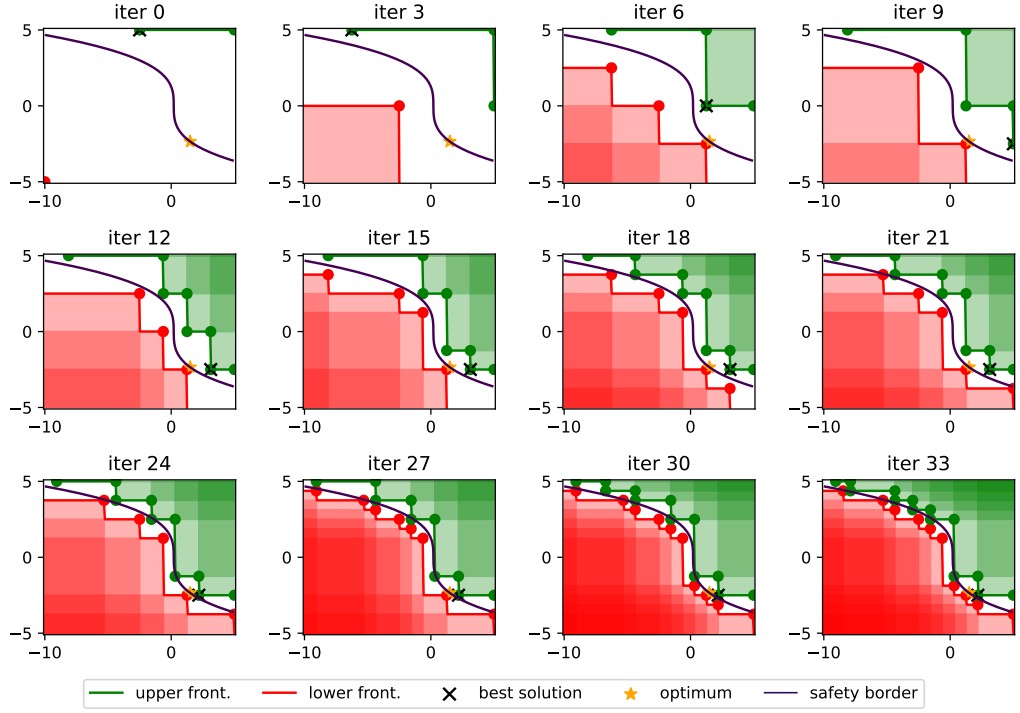

Figure 15: Frontier Search on the monotone optimization problem in in Eq. 57. Algorithm 1 quickly shrinks the area between the frontiers and returns solutions close to the optimum after only a handful of queries.

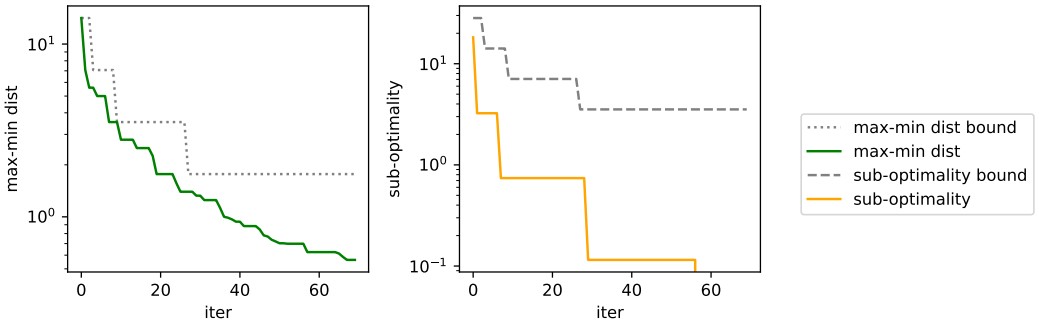

Figure 16: Convergence of Frontier Search for the optimization problem in Eq. 57.

| $n$ | $T$ | Duration Frontier Search | Duration F-PACOH |
|-----|-----|-------------------------|------------------|
| 10 | 50 | $14.19 \pm 0.52$ | $169.55 \pm 1.99$ |
| 10 | 100 | $18.83 \pm 0.62$ | $183.55 \pm 2.21$ |
| 10 | 200 | $33.82 \pm 1.26$ | $230.90 \pm 4.57$ |
| 10 | 400 | $75.23 \pm 10.42$ | $247.89 \pm 11.05$ |
| 20 | 50 | $13.96 \pm 0.87$ | $178.47 \pm 4.93$ |
| 20 | 100 | $22.14 \pm 0.68$ | $187.57 \pm 1.84$ |
| 20 | 200 | $37.44 \pm 2.54$ | $224.54 \pm 8.75$ |
| 20 | 400 | $95.15 \pm 11.65$ | $243.12 \pm 8.94$ |

Table 3: Compute time in seconds for Frontier Search and F-PACOH for different number of tasks $n$ and samples per task $T$. Reported is the mean and standard deviation over 5 sees / repetitions.

of sub-sequences of each dataset, as well as the maximum size of a sub-sequence used for GP inference grows. While the former can be parallelized, GP inference cannot be easily parallelized. The meta-training of the GP priors, typically takes ca. 3-4 minutes. Finally, the compute time for the safe BO part strongly varies depending on the employed algorithm (e.g. GoOSE or SafeOpt) and how time intensive/costly it is to query the $f$ and $q$. In the best case, when we use simulated toy functions that are very cheap to evaluate and run GoOSE for 200 steps, the safe BO takes ca. 500 seconds (8-9 min). In case of SafeOpt, the compute time is highly variable depending on the expander computation and ranges from 15 minutes to 1.5 hours. Generally, when employed with a real system or expensive simulation, the time for the safe BO outweights the compute time for the frontier search and F-PACOH steps of SaMBO by a order of magnitude.

### G.3 Study on amount of meta-train data

In this section, we aim to study how varying amounts of meta-training data affect the overall performance of SaMBO. For this purpose, we employ SaMBO-G on the the Camelback Function + Random Sinusoid environment with a varying number of meta-training tasks and data points per task. Figure 17 displays the terminal inference regret at ($T = 100$) for a varying number of tasks $n$ with each $m = 100$ data points (left) and a varying number of tasks $n$ with each $m = 100$ data points (right).

In all cases, the performance of SaMBO-G is substantially better than the inference regret of $> 0.1$ we obtain when running GoOSE with a Vanilla GP, as done in Figure 4. This demonstrates that we can also successfully perform positive transfer when the amount of meta-training data is smaller. Finally, we observe that the performance improvement we gain by doubling the number of tasks is much bigger than when we double the number of points per task. This is consistent with learning theory in e.g. Pentina and Lampert [30] and Rothfuss et al. [34].

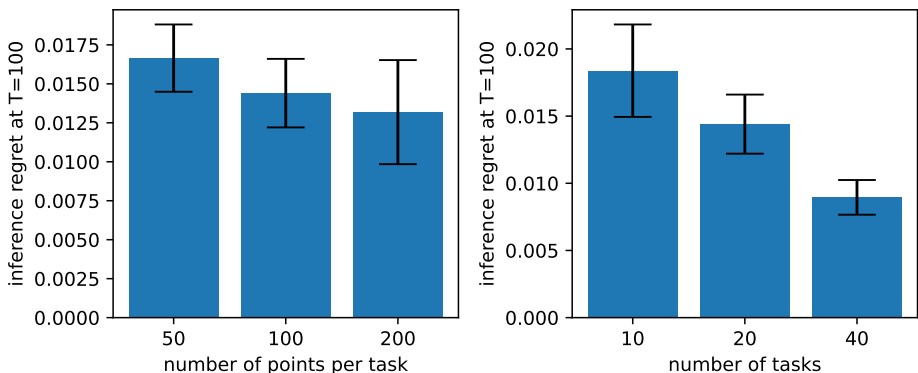

Figure 17: Overall performance for different amounts of meta-training. Displayed is the inference regret at last step ($T = 100$) of SaMBO-G in the Camelback Function + Random Sinusoid environment. Left: Varying number of data points per task $m$ for $n = 20$ tasks. Right: varying number of tasks $n$ with each $m = 100$ data points. Doubling the number of tasks leads to more performance improvements than doubling the number of points per tasks.

