# OpenReview forum: "Meta-Learning Priors for Safe Bayesian Optimization"
_robot-learning.org/CoRL/2022/Conference — CoRL 2022 Oral_

### Official Review · Reviewer_wnHy · 2022-07-26

**Originality:** Very Good
**Technical Quality:** Very Good
**Clarity Of Presentation:** Very Good
**Impact:** 4

**Recommendation:**

Weak Accept: I recommend accepting the paper, but will not argue for my recommendation if the majority of other reviewers have a different opinion.

**Summary:**

This paper presents a meta-learning method for model selection in the safe BO setting. The method selects the hyperparameters of the target and constraint GP models such that the BO procedure respects constraints while avoiding too conservative hyperparameters.

This is achieved by the maximizing calibration and sharpness of the uncertainty estimates over a set of datasets collected from related tasks.

The best hyperparameters are employed in a meta-learning procedure (F-PACOH) to learn informative and safe priors for the GP models used in BO with the goal of improving efficiency. This approach mitigates the risk of overconfident priors when data are scarce.
Once the GP priors are meta-learned, they can be employed for efficient Safe BO.

The approach demonstrates improved query efficiency with respect to baselines and respects constraints. Experiments are carried out in simulation in low-dimensions (2 benchmark functions and a PID tuning problem).


**Issues:**

Consider addressing the weaknesses listed above, i.e.,
- Comment upon scaling to higher-dimensional settings (or even show an experimental stress-test if time permits)
- Comment and provide examples of potential real-world scenarios in which similar tasks may typically be available or could be conveniently acquired, and how to quantify if a task is “similar enough” to another one for it to be employed for meta learning
- Discuss/show counter-examples and edge-cases for the monotonicity of the average calibration w.r.t. the kernel length scale
- Notation consistency between text and Algorithm environments
- Add short description of avg-std
- Fix the listed typos


**Quality Of The Limitations Section:**

Limitations are addressed clearly

**Reviewer Expertise:**

4: The reviewer is confident but not absolutely certain that the evaluation is correct

**Robotics Focus:**

Highly relevant to robotics but no hardware experiments

**Strengths And Weaknesses:**

The problem of selecting hyperparameters for safe BO so as to ensure constraint compliance while at the same time avoiding excessively conservative exploration is a highly relevant one, since it can hamper actual applications of Safe BO to parameter tuning for safety-critical and costly-to-operate systems. This paper represents a significant step forward to tackle this problem, providing means to reliably estimate hyperparameters and informative GP priors in the case in which related task data are available.

The paper elegantly and coherently brings together tools and methods from model calibration and meta-learning, presenting them in a coherent way, and proposes a novel frontier search algorithm to select safe hyperparameters for GP priors, leveraging monotonicity of the calibration frequency (heuristically) and sharpness measures.

From the technical point of view, the presented method seems sound. The improvement in terms of query efficiency is shown both experimentally and in Theorem 1.

The frontier search algorithm assumes monotonicity of the calibration frequency in the kernel length scale, which the authors state is a reasonable empirical heuristic in most cases. Still, it would be informative to elaborate more on possible edge cases and counterexamples.

A more in-depth discussion on the applicability of the method in real-world scenarios would also be beneficial. In particular, is it common to have access to offline data from similar tasks on a real system like a PID-controlled robot? How can we assess if the target task can be considered to belong to the same meta-distribution as the previously-collected data?

Experiments are carried out in low-dimensional parameter spaces, i.e., two 2-D benchmark functions and a 3-D PID gain tuning problem. The latter is not carried out on a physical system, and it is not clear how SaMBO would scale to higher dimensions. Yet, it represents a promising demonstration of the capabilities of SaMBO for many potential robotics and automation applications. The number of repetitions of the experiments appears adequate.

The quality of presentation is high. The paper is well-structured. Concepts are introduced at the right granularity and notation is not too heavy. Figures are informative and successful in clarifying the rationale of the frontier search algorithm and presenting the experimental results.

Few comments on potential improvements:
-	Consider harmonizing the notation in the figures and text, which appear to vary slightly in some cases (e.g., $Q_l$ vs. $Q^l$, define $f^{tar}$, $q^{tar}$)
-	I may be wrong, but, in Algorithm 1, wouldn’t the Prune function need to take as input the intersection between Q and z instead of the union?
-	The definition of average predictive std in Eq. (6) may be worth an additional short description in a way similar to Eq. (5) to better appreciate the conceptual difference

Others:

- L36: on which it falls back to $\rightarrow$  on which it falls back
- L37: data-riven $\rightarrow$ data-driven
- L51: GOOSE [10, 18] by expanding $\rightarrow$ GOOSE [10, 18] does so by expanding
- L60: non-i.i.d $\rightarrow$ non-i.i.d.
- L61: settings, however, $\rightarrow$ settings. However,
- L77: e.g., data of the same robotic platform under different conditions. $\rightarrow$  For example?
- L85: we ask the question how $\rightarrow$ we ask the question of how
- L93 where $\beta$ *is* the… *and* is set such…
- L100: w.r.t. $\rightarrow$  with respect to
- L114: Prior work *proposing* to *meta-learn* GP priors [44, 45, 32], *though*, fail…
- L117, missing a closing parenthesis in the inline expression
- Consider using the definition symbol in Eqq. (2), (3), (4), (5), (6)
- Add full stop after Eq. (4)
- Consider redefining “calib” by “avg-calib” in Eq. (5), since in the tect (L147) it is referred to as average calibration frequency and it is averaged in a similar way as “avg-std”
- L229 calibration- and sharpness-based
- L259 to measure a method’s
- L287 Note that, in case of the regret, the …

**Summary Of Recommendation:**

Overall, the paper presents a novel method tackling a well-known relevant problem in Safe BO. To the best of my knowledge, relevant literature is exhaustively covered. Presentation quality is good. The method employs tools from calibration analysis and meta learning in an innovative way, and presents a novel frontier search method to safely select informative priors and improve query efficiency of BO procedures when data from similar tasks is available.

The efficiency and safety of the method is demonstrated with two different BO algorithms on three low-dimensional tasks in simulation. Although not demonstrated on real hardware, the results seem promising and the method may be applied to a wide range of robot control and automation problems in which gains need be tuned under strict safety constraints. Scaling properties of this approach to higher dimensions remain to be investigated.

In my view, the presented method and results are timely and relevant to the robot learning community.

---

> ### Author Response · Authors · 2022-08-26
> **Rebuttal response**
>
> Thanks a lot for going through our manuscript in so much detail. We have fixed the typos and implemented the small tips for improvement that you have pointed out. We are happy to hear that you liked the paper and recommend acceptance. In the following we address your questions and suggestions:
>
> **Comment upon scaling to higher-dimensional settings.**
>
> Thank you for raising this excellent point. Similar to other BO / safe BO methods that are based on GPs, the sample complexity grows exponentially with the number of dimensions. The meta-learning in our case alleviates this issue to some degree by making the GP prior more informed about our environment of tasks in areas where meta-training data is available. However, to maintain safety, we regularize the meta-learned GP prior towards a Vanilla GP with SE kernel (cf. Eq. 3) in areas without or with little meta-training data.
> The higher-dimensional our safe BO problem, the sparser is the meta-training data in the domain. Thus, the areas where our meta-learned GP resembles a Vanilla GP become larger in proportion and we face again the general issue of poor sample complexity in high dimensions. Generally, without the additional assumption that, e.g., there exists some lower-dimensional sub-space in which the functions we are optimizing lie, we do not believe that the ‘curse of dimensionality problem can be solved while assuring safety.
> In response to your suggestion, we have added a section to the Appendix (now Appendix E) where we discuss the limitations of SaMBO to higher-dimensional problems.
>
> **I may be wrong, but, in Algorithm 1, wouldn’t the Prune function need to take as input the intersection between Q and z instead of the union?**
>
> The prune function in Algorithm 1 correctly takes as an input the union of Q and z. Given the the union Q and z, it removes points from Q that are dominated by z.
>
> **Consider redefining “calib” by “avg-calib” in Eq. (5), since in the tect (L147) it is referred to as average calibration frequency and it is averaged in a similar way as “avg-std”**
>
> We are grateful for this suggestion, which makes a lot of sense. We have renamed “calib” to “avg-calib”.
>
> **Discuss/show counter-examples and edge-cases for the monotonicity of the average calibration w.r.t. the kernel length scale**
>
> We have added a detailed discussion on the monototonicity properties of the calibration- and sharpness-based optimization problem to Appendix E. There, we explain under what circumstances, i.e. when the kernel variance $\nu$ is smaller than the variance of the data, and why the monotonicity of the calibration frequency no longer holds. We also added a Figure that illustrates the monotonicity for $\nu > 1$ and a breakdown thereof for $\nu < 1$. Moreover, we explain how we can easily avoid issues due to non-monotonicity by setting the lower bound of the frontier search for the kernel variance to the data variance (or 1 if the data has been standardized).
>
> **Add short description of avg-std**
>
> We thank the reviewer for this suggestion. We have added a sentence below Eq. 6 which conceptually explains the avg-std

---

> > ### Comment · Reviewer_wnHy · 2022-08-27
> > **Acknowledgment of Authors' Response**
> >
> > I would like to thank the authors for their detailed responses and improvements to the paper. They satisfactorily answer my questions and doubts. I confirm my positive score supporting acceptance.
> >
> > I suggest to split and explicitly refer to the new Appendix E subsections in the main text. That is, to Appx. E.1 (scaling to high dimensions) in the limitations section and to Appx. E.2 (monotonicity) in Section 4.2.
> >
> > Also consider to double-check whether all other relevant Appendix sections are referred to by the main text.

---

### Official Review · Reviewer_iFMD · 2022-07-26

**Originality:** Good
**Technical Quality:** Excellent
**Clarity Of Presentation:** Very Good
**Impact:** 3

**Recommendation:**

Strong Accept: I recommend accepting the paper and will argue for my recommendation even if other reviewers hold a different opinion.

**Summary:**

The authors present a method for meta-learning priors for safe bayesian optimization (BO) from data. The authors contribute a method for finding safety-compliant kernel hyper-parameters by developing an optimization problem which maximizes sharpness while maintaining calibration at all times. By exploiting monotonicity properties of the optimization problem, the authors develop an efficient search algorithm, “Frontier Search” to converge to optimal hyper-parameters. They contribute a safe BO algorithm “SaMBO”, which uses the frontier search in combination with F-PACOH and existing safe BO methods. The authors show that their method requires fewer iterations to converge as compared with data-free methods.

**Issues:**

Content issues/questions:
- Related work: I believe there are a number of related works in the area of PAC-bayes meta-learning regularization which could be added. E.g.: Pentane & Lampert NeurIPS 2015, Amit & Mier ICML 2018, Balaji et al. NeurIPS, 2018, Yin et al. ICLR 2020, Farid & Majumdar NeurIPS 2022. I think meta-regularization would be fair to mention as it relates to both uncertainty estimation and PAC-Bayes meta-learning which are already discussed in the related work.

- L137 What exactly about eq. (5) and (6) are different from [47, 48, 49, 50]? If the authors reference [47-50] purely to refer the reader to information about calibration and sharpness, it would be more clear to say “see e.g. [47, 48, 49, 50]”.

- L167: The assumption that the optimal (and even safe) kernel hyper-parameters have known upper and lower bounds is a key assumption used to prove Lemma 1 and Theorem 1. However, I do not see this assumption as well motivated/argued. Why can we assume that we known a region of safe hyper-parameter? If we choose a very large range to ensure the box includes the optimal solution, is that damaging to the sample efficiency of the Frontier Search?

- Lemma 1, Theorem 2: I would appreciate a proof outline/intuition in the body of the paper. At least add a reference to the proofs in the appendix so the reader can find them more easily.

- Section 6 Experiments: I think it is important to report computation of the proposed method SaMBO in comparison to methods which do not use the frontier search. The authors mention that Equations (6) and (7) are expensive. Since this is additional computation not usually performed in Safe BO methods, it is not clear to me that one is preferable in terms of total processing time.

- L253: It is interesting that the method works with non-i.i.d. within-task data. Is there a significant improvement in performance if the method is provided with i.i.d data? If there is not a significant difference, this would be a major positive of the approach.

- L265: Do the authors have any intuition as to why log-space of l and v work better? It would be interesting to see a comparison of log and linear search.

- Figure 3: Which hyper-parameter selection do the authors use for GoOSE/SafeOpt? Can the hyper-parameters be tuned in any (potentially naive) way by manually looking at the reference data you give to SaMBO-G/S to provide a fairer comparison.

- L302 “our framework makes safe BO free of hyper-parameters”. I believe $z_1^l, z_1^u, z_2^l, z_2^u$ (L167-L168) should be considered hyper-parameters in addition to the standard deviation of the Gaussian likelihood of the GP (L751). I agree that the proposed method is more practical in general, but it is not correct to say that SaMBO is free of hyper-parameters.

- L302-L305: I believe the authors could expand on the discussion of the limitations. Can discuss possible time to compute Equations (5) and (6), are there cases where (5) and (6) are intractable? Similarly, are there cases where specifying l and v are straightforward, and therefore the frontier search methods is not necessary? Also, no longer needing to specify l and v moves the problem of hyper-parameter specification for Safe BO to other parameters — are there cases in which these other parameters are just as difficult to specify?

Editorial issues:
- Handful of minor typos: L10 and L37: “data-riven” -> “data-driven”, L51: “GoOSE by expanding” -> “GoOSE explores by expanding”, L234 “fist” -> “first
- I am not certain the authors are using “c.f.” correctly. E.g. in L169 “Since both s(z) and c(z) are monotone we know that z* must lie on or directly above the constraint boundary c(z) = 1 (c.f. Lemma 2).”. I would interpret “(c.f. Lemma 2)” as “compared to Lemma 2”, but it seems that the authors mean “see Lemma 2”. Indeed, Lemma 2 states that “z* lies on or directly above the constraint boundary”. I believe the authors make this mistake on L194 and L762 as well.

**Quality Of The Limitations Section:**

Additional details required

**Reviewer Expertise:**

3: The reviewer is fairly confident that the evaluation is correct

**Robotics Focus:**

Highly relevant to robotics but no hardware experiments

**Strengths And Weaknesses:**

Strengths
- Section 3 problem statement and background is informative and clear. It introduces the background thoroughly and in an accessible way.
- The methods and contributions are well presented.
- The contribution is useful, and as far as I can tell, novel. The proposed method has the potential to significantly reduce computation time for safe BO without significantly complicating implementation.
- Results (Fig 3, Fig 4) show clear improvement in number of iterations required in BO as compared with baseline, data-free, methods.

Weaknesses
- If I am understanding the examples correctly, you only compare SaMBO-G/S to the data-free baselines. It would be important to compare SaMBO to methods which can use the calibration data which SaMBO uses.
- The authors make safe BO more efficient by spending computation time on the Frontier Search. It’s not immediately clear that the overall result is more efficient. Nor are there any results which discuss or compare overall efficiency (in terms of e.g. wall-clock time of the entire BO algorithm including the Frontier Search).
- Some assumptions are not entirely clear or well justified. E.g. the authors say that safe BO methods assume that correct kernel hyper-parameters are given (L52). However, their assumption that upper and lower bounds on the correct hyper-parameters (L167) is not well justified.

**Summary Of Recommendation:**

The challenge of hyper-parameter selection is ubiquitous and the authors present a practical, data-based, method for selecting kernel hyper-parameters. When combined with existing safe BO methods, the authors show their method requires fewer iterations to converge when compared with data-free methods. However, it is not entirely clear that the method has better computation time (i.e. in terms of wall-clock time) since no results for this are presented. I would also like to see a fairer comparison (i.e. to methods which also use data for calibration) such as a hand-tuned baseline or naive approach. However, as far as I am aware, no other method for this kind of safe kernel hyper-parameter selection exists. The methods are convincing and well presented. Therefore, I recommend weak acceptance.

---

**Update**: After reading the replies, the authors have clarified or improved upon all major issues and weaknesses. I have increased my recommendation to strong accept.

---

> ### Author Response · Authors · 2022-08-26
> **Rebuttal response 1/2**
>
> Thanks a lot for the detailed and encouraging feedback as well as your suggestions on how to improve the paper. In the following, we address your points and questions in detail:
>
> **If I am understanding the examples correctly, you only compare SaMBO-G/S to the data-free baselines. It would be important to compare SaMBO to methods which can use the calibration data which SaMBO uses.**
>
> In our experiment, we actually do what you suggest. We compare SaMBO-G/S to GoOSE and SafeOpt with GPs where the kernel parameters have been selected with the calibration/sharpness-based frontier search from Section 4. This is mentioned in line 285 - 286, although very briefly due to the space constraints. Hence, all methods reported in Figure 3 and 4 make use of the meta-training data. In case of GoOSE and SafeOpt, the data is used for finding kernel parameters. In case of SaMBO-G/S, the data is first used for finding the kernel parameters and then for meta-training the GP prior.
>
> **[...] discuss possible time to compute Equations (5) and (6), [...]**
>
> Thank you for the suggestion. We have added a discussion to Appendix D how to efficiently compute the calibration and sharpness in equation (5) and (6) using parallelization over tasks and sub-sequences. This allows us to perform the frontier search reasonably fast (e.g. for 20 tasks and 200 samples per tasks in ca. 30 seconds). In addition, we have added an empirical compute time analysis to Appendix G.
>
> Generally, the fundamental premise of safe BO and BO more broadly is that we want to optimize a black-box function that is costly to query, e.g. running the robot with a certain controller configuration and measuring relevant metrics such as the tracking error. In most cases, function evaluations, i.e. running the robot, do not only take time but also cost energy and cause wear and tear. In nearly all practical settings, the computational cost of the frontier search + F-PACOH is a small fraction of the time it takes to run safe BO (for e.g. 200 steps) on the real system or a high-fidelity simulation thereof.
>
> **[..] are there cases where specifying l and v are straightforward, and therefore the frontier search methods is not necessary?**
>
> The lengthscale is fundamentally related to the somoothness/variability of the functions we want to fit. In prcactice, we rarely know how variable the true underlying function is. We do not know any practical application where the ‘correct’ lengthscale parameter is naturally given or specified.
>
> **Related work: I believe there are a number of related works in the area of PAC-bayes meta-learning regularization which could be added. E.g.: Pentane & Lampert NeurIPS 2015, Amit & Mier ICML 2018, Balaji et al. NeurIPS, 2018, Yin et al. ICLR 2020, Farid & Majumdar NeurIPS 2022. I think meta-regularization would be fair to mention as it relates to both uncertainty estimation and PAC-Bayes meta-learning which are already discussed in the related work.**
>
> We thank for the suggested related works and agree that they are relevant. Thus, we have incorporated the suggested related work on meta-regularization into the related work section.

---

> > ### Comment · Reviewer_iFMD · 2022-08-26
> > **Reply to Paper252 Authors**
> >
> > Thanks very much for your detailed response. I appreciate the precisions in addressing my comments.
> >
> > The explanations have clarified my understanding of the results and method, and they are stronger than I originally thought. In particular that GoOSE and SafeOpt has been calibrated in the same way as SaMBO-G/S. In addition, the note that the data is standardized and therefore the choices of domain boundaries is fairly straightforward.
> > I also appreciate the additional results on compute time as it provides a better perspective on the time of the frontier search in comparison to F-PACOH.

---

> ### Author Response · Authors · 2022-08-26
> **Rebuttal response 2/2**
>
> **[...] the authors say that safe BO methods assume that correct kernel hyper-parameters are given (L52). However, their assumption that upper and lower bounds on the correct hyper-parameters (L167) is not well justified.**
>
> As usual in machine learning, we standardize the data so that is has approximately mean 0 and variance 1. This way, data with originally different value ranges are projected into the same range. Now that we work in a standardized value range, it it fairly easy to choose a reasonable upper and lower bound for the kernel variance and lenthscale. For instance, for the lengthscale we use 0.01 (very high variation) as a lower bound and 5.0 (very low variation) as an upper bound. Since the frontier search converges with a relatively fast rate, choosing the interval conservatively as we do suffices and does not really play a major role for the practical performance. In our experiments, 20 iterations of frontier search with the relatively large ranges stated above suffice to find a good solution.
>
> Moreover, if, for some reason, the value range of the kernel parameters should not be large enough, the frontier search returns a value on the boundary of its domain which is very easy to diagnose and fix by enlarging the search domain in the corresponding direction.
>
> In summary:
> * In the standardized space it is easy to choose reasonable domain boundaries.
> * Due to the fast convergence of frontier search, they only play a minor role.
> * Domain boundaries that are too narrow are very easy to diagnose and fix in practice.
>
> Hence, having valid search domain boundaries for the kernel parameters is a weak assumption which is easy to fulfill in practice and has only minor implications on the performance. In contrast, choosing the kernel parameters directly is difficult and has major implication on the downstream data-efficiency and safety.
>
> **[...] report computation of the proposed method SaMBO [...]. The authors mention that Equations (6) and (7) are expensive. Since this is additional computation not usually performed in Safe BO methods, it is not clear to me that one is preferable in terms of total processing time.**
>
> We have added a discussion to Appendix D how to efficiently compute the calibration and sharpness in equation (5) and (6) using parallelization over tasks and sub-sequences. This allows us to perform the frontier search reasonably fast (e.g. for 20 tasks and 200 samples per tasks in ca. 30 seconds). In response to your suggestions, we have added an empirical compute time analysis to Appendix G.
>
> Generally, the fundamental premise of safe BO and BO more broadly is that we want to optimize a black-box function that is costly to query, e.g. running the robot with a certain controller configuration and measuring relevant metrics such as the tracking error. In most cases, function evaluations, i.e. running the robot, do not only take time but also cost energy and cause wear and tear. In nearly all practical settings, the computational cost of the frontier search + F-PACOH is a small fraction of the time it takes to run safe BO (for e.g. 200 steps) on the real system or a high-fidelity simulation thereof.
>
> **L253: It is interesting that the method works with non-i.i.d. within-task data. Is there a significant improvement in performance if the method is provided with i.i.d data? If there is not a significant difference, this would be a major positive of the approach.
> We thank the reviewer for raising this question.**
>
> The ‘class’ of non-i.i.d. data is huge and incorporates everything from negatively dependent data (usually better than i.i.d. for learning) to strongly positively correlated data (usually worse to learn with). Thus, we cannot give a general answer to this question.
> However, in our experiments we tried our method with 1) data uniformly sampled from the safe region (i.i.d) and 2) data collected with SafeOpt (non-i.i.d). We did not observe a significant performance difference between to two ways of constructing the datasets. Since using data collected via SafeOpt is more realistic, we decided to report the experimental results with the non-i.i.d. data.
>
> **Typos and re-wording suggestions:**
> Thanks a lot for reading the manuscript so thoroughly and pointing out individual typos. We have fixed them in the updated manuscript

---

### Official Review · Reviewer_SuCp · 2022-08-01

**Originality:** Very Good
**Technical Quality:** Excellent
**Clarity Of Presentation:** Excellent
**Impact:** 3

**Recommendation:**

Strong Accept: I recommend accepting the paper and will argue for my recommendation even if other reviewers hold a different opinion.

**Summary:**

In this paper, the authors present an improved safe Bayesian optimization method by exploiting existing offline data for the meta-learning of priors. By introducing safety-compliant priors via empirical uncertainty metrics and a frontier search algorithm, the proposed framework provides a reliable uncertainty quantification which leads to more trustworthy safety guarantees in contrast to hand-tuned vanilla safe BO. A simulation and a robotic experiment show the benefit of the framework.

**Issues:**

Please address the cons mentioned above.

**Quality Of The Limitations Section:**

Limitations are addressed clearly

**Reviewer Expertise:**

3: The reviewer is fairly confident that the evaluation is correct

**Robotics Focus:**

Sufficient demonstration on hardware

**Strengths And Weaknesses:**

Pros:
+ The paper is very well written and clear in all its steps
+ The topic is of timely interest and important for the robotics community
+ The method allows to exploit eventually existing offline data.
+ The experiment/simulation are convincing

Cons:
- The footnote "the monotonicity of the calibration frequency in l is only an empirical heuristic that holds in almost all cases if ν is at least as big as the variance of the targets y in a dataset" seems to be quite important as the monotonicity is key for the success of the optimization. I would like to see some more discussion and evaluations here.

- The proposed framework seems to be computational expensive. Some more details on that would be of interest.

**Summary Of Recommendation:**

I really enjoyed reading this paper as it is well written and technical sound. The idea is interesting and can be definitely helpful in scenarios where offline-data is existing. However, I'm not sure if such a rich offline-data set is often available such that a major impact in robotics is questionable.

---

> ### Author Response · Authors · 2022-08-26
> **Rebuttal response**
>
> Thanks a lot for the encouraging feedback and the tips for improving the paper. We have updated the manuscript based on them. In the following we provide more details on how:
>
> **The footnote "the monotonicity of the calibration frequency in l is only an empirical heuristic that holds in almost all cases if ν is at least as big as the variance of the targets y in a dataset" seems to be quite important as the monotonicity is key for the success of the optimization. I would like to see some more discussion and evaluations here.**
>
> We agree that we have discussed the monotonicity of (7) insufficiently. For this reason, we have added a detailed discussion on the monotonicity properties of the calibration- and sharpness-based optimization problem to Appendix D. There, we explain under what circumstances, i.e. when the kernel variance $\nu$ is smaller than the variance of the data, and why the monotonicity of the calibration frequency no longer holds. We also added a Figure that illustrates the monotonicity for $\nu > 1$ and a breakdown thereof for $\nu < 1$. Moreover, we explain how we can easily avoid issues due to non-monotonicity by setting the lower bound of the frontier search for the kernel variance to the data variance (or 1 if the data has been standardized).
>
> **The proposed framework seems to be computationally expensive. Some more details on that would be of interest.**
>
> Thank you for the suggestion. We have added a discussion to Appendix D how to efficiently compute the calibration and sharpness in equation (5) and (6) using parallelization over tasks and sub-sequences. This allows us to perform the frontier search reasonably fast (e.g. for 20 tasks and 200 samples per tasks in ca. 30 seconds). In addition, we have added an empirical compute time analysis to Appendix G.

---

> > ### Comment · Reviewer_SuCp · 2022-08-26
> > **Reviewer Response**
> >
> > Thank you for addressing all of my concerns. I appreciate your work!

---

### Official Review · Reviewer_jWPh · 2022-08-01

**Originality:** Good
**Technical Quality:** Very Good
**Clarity Of Presentation:** Fair
**Impact:** 3

**Recommendation:**

Weak Accept: I recommend accepting the paper, but will not argue for my recommendation if the majority of other reviewers have a different opinion.

**Summary:**

This paper presents a meta-learning framework to optimize controllers with safety constraints with bayesian optimization. To this end, the authors build on recent work for tuning gaussian process parameters and kernels with meta-learning. The authors test this approach on several bayesian optimization benchmarks and for tuning a controller for a linear robot system. The results demonstrate how optimization with meta-learning priors converges faster than previous paradigms while satisfying safety constraints.

**Issues:**

Section 1.
- F-PACOH is an acronym that is very important to this paper, yet the authors do not describe its meaning explicitly.
Section 4.2.
- The frontier search is hard to follow over long paragraphs and can, at times, be confusing to the reader.
- the functions LARGESTMAXMINRECT() and BESTWORSTCASEQUERY() are not defined in the paper, is up to the reader to infer their meaning.
- It would be much more helpful if the authors described the functions LARGESTMAXMINRECT() and BESTWORSTCASEQUERY() under an equation or a separate algorithm environment.
- If Theorem 1 is not proven in the main paper it is very hard to assess its validity or impact.
Section 6.1.
- Why is regret the best evaluation metric? It would be helpful to comment on this
Section 7.
- What are the limitations of this method when applied to larger problems?
- What kind of BO problems are hard to represent with the F-PACOH modeling?
- What are future lines of work where this algorithm could be impactful?

**Quality Of The Limitations Section:**

Additional details required

**Reviewer Expertise:**

2: The reviewer is willing to defend the evaluation, but it is quite likely that the reviewer did not understand central parts of the paper

**Robotics Focus:**

Relevant but unlikely to deploy to hardware in near future

**Strengths And Weaknesses:**

Strengths:

1. The paper offers a strong technical analysis of different meta-learning paradigms for gaussian-processes
2. The paper runs thorough experiments to demonstrate the capabilities of this approach on several benchmarks.

Weaknesses:

1. The limitations section of this approach does not discuss algorithmic complexity or implications when applied to larger optimization problems.
2. Theorems and Lemmas are presented but not proven in the main paper, which forces the reader to look over appendices.
3. Several acronyms are used repeatedly over the paper and not explicitly defined. The presentation could be improved.

**Summary Of Recommendation:**

The paper offers an interesting contribution and is written in a fashion that walks the unfamiliar reader through the key concepts that build this algorithm. The results are impressive and seem like a helpful contribution to the field of meta-learning for robotics. I recommend acceptance, but the authors can significantly improve the paper by improving the presentation and clarifying on the limitations of the approach.

---

> ### Author Response · Authors · 2022-08-26
> **Rebuttal response**
>
> Thank you for engaging with our paper and giving useful tips on how to improve it. In the following, we answer to your questions / points:
>
> **[...] the functions LARGESTMAXMINRECT() and BESTWORSTCASEQUERY() are not defined in the paper, is up to the reader to infer their meaning.**
>
> The functions LARGESTMAXMINRECT() and BESTWORSTCASEQUERY() are both formally defined in Appendix C (Definition 5 and Equation 43). We agree if the reviewer that the FrontierSearch algorithm is harder to follow since relevant parts are deferred to Appendix. Unfortunately, due to the page limit of 8 pages, we could not define them in the main text. We now reference to Appendix C in the main text, so that readers can at least find the relevant definitions more easily.
>
> **Why is regret the best evaluation metric? It would be helpful to comment on this Section 7.**
> The safe regret measures how far the current best pick of the safe BO algorithm is away from the safe optimum. It is a metric that is easy to interpret since values close to 0 mean that the safe BO algorithm has found a solution close to the safe optimum. It is a common metric in the field of safe BO that is, e.g. used in [1, 2]. Because of space limitations, we have not included this discussion.
>
> **What are the limitations of this method when applied to larger problems?**
> We thank the reviewer for raising this point. Similar to other BO / safe BO methods that are based on GPs, the sample complexity grows exponentially with the number of dimensions. The meta-learning in our case alleviates this issue to some degree by making the GP prior more informed about our environment of tasks in areas where meta-training data is available. However, to maintain safety, we regularize the meta-learned GP prior towards a Vanilla GP with SE kernel (cf. Eq. 3) in areas without or with little meta-training data.
>
> The higher-dimensional our safe BO problem, the sparser is the meta-training data in the domain. Thus, the areas where our meta-learned GP resembles a Vanilla GP become larger in proportion and we face again the general issue of poor sample complexity in high dimensions. Generally, without the additional assumption that, e.g., there exists some lower-dimensional sub-space in which the functions we are optimizing lie, we do not believe that the ‘curse of dimensionality problem can be solved while assuring safety.
> In response to your questions, we have added a section to the Appendix (now Appendix E) where we discuss the limitations of SaMBO to higher-dimensional problems.
>
> **What are future lines of work where this algorithm could be impactful?**
> We thank the reviewer for this interesting question. Generally, there is a broad range of work that relies on calibrated uncertainty estimates of an underlying probabilistic regression model. The great majority of this work assumes that a calibrated model is simply provided (e.g. in form of a GP with known kernel parameters) which is not the case in practice. Our proposed method provides a data-driven solution to this problem of how to obtain a calibrated, but yet sharp, probabilistic regression model. We demonstrate its effectiveness in safe BO. However, further lines of work such as principled exploration in model-based RL (e.g. [3]) or safe RL (e.g. [4, 5]) can greatly benefit from our algorithm. In a similar way, work on offline RL / off-policy evaluation such as [6] is amenable to our approach.
>
> [1] Turchetta, M., Berkenkamp, F., & Krause, A. (2019). Safe exploration for interactive machine learning. Advances in Neural Information Processing Systems, 32.
>
> [2] Amani, S., Alizadeh, M., & Thrampoulidis, C. (2020). Regret bound for safe gaussian process bandit optimization. In Learning for Dynamics and Control (pp. 158-159). PMLR.
>
> [3] Chowdhury, S. R., & Gopalan, A. (2019, April). Online learning in kernelized markov decision processes. In The 22nd International Conference on Artificial Intelligence and Statistics (pp. 3197-3205). PMLR.
>
> [4] Berkenkamp, F., Turchetta, M., Schoellig, A., & Krause, A. (2017). Safe model-based reinforcement learning with stability guarantees. Advances in neural information processing systems, 30.
>
> [5] Wachi, A., Sui, Y., Yue, Y., & Ono, M. (2018, April). Safe exploration and optimization of constrained mdps using gaussian processes. In Proceedings of the AAAI Conference on Artificial Intelligence (Vol. 32, No. 1).
>
> [6] Yu, T., Thomas, G., Yu, L., Ermon, S., Zou, J. Y., Levine, S., ... & Ma, T. (2020). Mopo: Model-based offline policy optimization. Advances in Neural Information Processing Systems, 33, 14129-14142.

---

### Author Response · Authors · 2022-08-26
**Updates to the paper in response to the reviewers inquiries and suggestions:**

In response to the reviews, we have made many smaller changes to the manuscript which are mentioned in our rebuttal responses.  Implementing the reviewers suggestions, we have made the following bigger edits and additions the updated manuscript:

* Additional related work on meta-regularization (Section 2)
* Better description of avg-std (Section 4.1)
* Explanation on how to efficiently compute the calibration and sharpness metrics in Eq. 5 and 6 via parallelization.
* Discussion on the applicability to higher-dimensional problems (Appendix F)
* Discussion on the monotonicity of the calibration-sharpness optimization problem (Appendix F)
* Compute time analysis and discussion (Appendix G2)
* Empirical study on the amount of meta-training data (Appendix G3)

Unfortunately, the page limit of 8 pages remains unchanged after the rebuttal. Thus, many additions had to go into the appendix instead of the main text. The updated manuscript is attached to this comment.

We thank the reviewers once more for their constructive feedback and hope that we were able to address most of the suggestions and concerns with these updates.

---

### Meta-Review · Area_Chair_jvyC · 2022-08-03

**Recommendation:** Accept (Oral)
**Confidence:** 4

**Metareview:**

Please check the comments of the reviewers in detail.

### Strengths
- the paper is very well written and clear
- the topic is important for robotics
- promising experimental results (in simulation)

### Weaknesses
- evaluation in 2 2-dimensional functions and 1 simulated robot (3-dimensional problem): most problems in robotics are higher dimensional
- no experiment with the real system (simulation only)
- more discussion is needed about the computational cost of the approach
- no discussion about how much data is needed for the meta-training (e.g., plot the final performance vs the amount of prior data), and how often/why this data would be available

(minor remark: it would be useful to squeeze a picture of the robotic system in the main paper).

### Post-rebuttal review
I would like to thank the authors for their efforts in answering the comments and improving the paper. All reviewers agree that this is a high-quality paper.